# Train for the Worst, Plan for the Best: Understanding Token Ordering in Masked Diffusions

**Jaeyeon Kim** [*1]  **Kulin Shah** [*2]  **Vasilis Kontonis** [2]  **Sham Kakade** [1]  **Sitan Chen** [1]

## Abstract

In recent years, masked diffusion models (MDMs) have emerged as a promising alternative approach for generative modeling over discrete domains. Compared to autoregressive models (ARMs), MDMs trade off complexity at training time with flexibility at inference time. At training time, they must learn to solve an exponentially large number of infilling problems, but at inference time, they can decode tokens in essentially arbitrary order. In this work, we closely examine these two competing effects. On the training front, we theoretically and empirically demonstrate that MDMs indeed train on computationally intractable subproblems compared to their autoregressive counterparts. On the inference front, we show that a suitable strategy for adaptively choosing the token decoding order significantly enhances the capabilities of MDMs, allowing them to sidestep hard subproblems. On logic puzzles like Sudoku, we show that adaptive inference can boost solving accuracy in pretrained MDMs from $< 7\%$ to $\approx 90\%$, even outperforming ARMs with $7\times$ as many parameters and that were explicitly trained via teacher forcing to learn the right order of decoding.

## 1. Introduction

While diffusion models (Ho et al., 2020; Song et al., 2021) are now the dominant approach for generative modeling in continuous domains like image, video, and audio, efforts to extend this methodology to discrete domains like text and proteins (Austin et al., 2021; Lou et al., 2024; Hoogeboom et al., 2021b) remain nascent. Among numerous proposals, masked diffusion models (MDMs) (Lou et al., 2024; Sahoo et al., 2025; Shi et al., 2024) have emerged as a leading

---
[*]Equal contribution  [1]Harvard University  [2]University of Texas Austin. Correspondence to: Kulin Shah <kulin-shah@utexas.edu>.

*Proceedings of the 42nd International Conference on Machine Learning*, Vancouver, Canada. PMLR 267, 2025. Copyright 2025 by the author(s).

variant, distinguished by a simple and principled objective: to generate samples, learn to reverse a noise process which independently and randomly masks tokens.

In many applications, such as language modeling, masked diffusion models (MDMs) still underperform compared to autoregressive models (ARMs) (Nie et al., 2024; Zheng et al., 2024), which instead learn to reverse a noise process that unmasks tokens sequentially from left to right. However, recent studies suggest that MDMs may offer advantages in areas where ARMs fall short, including reasoning (Nie et al., 2024; Kitouni et al., 2025), planning (Ye et al., 2024), and infilling (Gong et al., 2024). This raises a key question: what are the strengths and limitations of MDMs compared to ARMs, and under what conditions can MDMs be scaled to challenge the dominance of ARMs in discrete generative modeling?

To understand these questions, we turn a microscope to two key competing factors when weighing the merits of MDMs over ARMs:

- **Complexity at training time**: By design, the prediction task that MDMs are trained on is more challenging. Whereas ARMs seek to predict the next token given an unmasked prefix, MDMs seek to predict a token conditioned on a set of unmasked tokens in arbitrary positions.

- **Flexibility at inference time**: On the other hand, the sampling paths taken by an MDM are less rigid. The order in which tokens are decoded at inference time is random instead of fixed to left-to-right. In fact, even more is possible: MDMs can actually be used to decode in *any order*.

Therefore, we ask:

*Are the benefits of inference flexibility for MDMs enough to outweigh the drawbacks of training complexity?*

In this work, we provide dual perspectives on this question.

**(1) Training for the worst.** First, we provide theoretical and empirical evidence that the overhead imposed by training complexity quantifiably impacts MDMs' performance.

We prove that even for simple, benign models of data, there are noise levels at which a large fraction, but not all, of

the corresponding subproblems solved by MDMs are computationally intractable. We then show this imbalance in computational complexity across subproblems persists even in real-world text data (Fig. 2, left).

**(2) Planning for the best.** While the above might appear to be bad news for MDMs, in the second part of this paper we answer our guiding question in the affirmative by building upon the observation (Zheng et al., 2023) that MDMs which can perfectly solve all masking subproblems can be used to decode in *any* order.

In place of vanilla MDM inference whereby tokens are unmasked in random order, we consider *adaptive* strategies that carefully select which token to unmask next. Our key insight is that this adaptivity makes it possible to *sidestep* the hard subproblems from training (Fig. 1). In fact, we find that **even without modifying how MDMs are trained, the resulting models' logits contain enough information to determine the right order in which to unmask.**

Our main empirical result is to show that the performance of MDMs pretrained on logic puzzle data dramatically improves when one goes from vanilla to adaptive inference. For example, on Sudoku puzzles, a simple adaptive strategy (Section 4.1) improves the accuracy of MDMs from $< 7\%$ to almost 90%. Remarkably, this not only outperforms vanilla ARMs, but even bespoke ARMs trained to learn the right decoding order via supervised teacher forcing (Shah et al., 2024; Lehnert et al., 2024) (Table 2).

**Organization.** In Section 2, we provide preliminaries on MDMs and set notation. In Section 3, we examine MDM training and demonstrate the imbalance in computational intractability across subproblems. In Section 4, we consider adaptive inference in MDMs and investigate its impact on likelihood modeling across various tasks.

## 2. Masked Diffusion Models (MDM)

In this section, we explain the framework of Masked Diffusion Models (Shi et al., 2024; Sahoo et al., 2025) and its interpretation as an *order-agnostic learner*. MDMs gradually add noise to the true discrete data and learn the marginal distribution of the induced reverse process. Below, we formulate the forward and reverse processes for MDMs.

Let the distribution $p_{\text{data}}$ on $\{1, \ldots, m\}^L$ be the data distribution over sequences of length $L$ and with vocabulary $\{1, \ldots, m\}$. We use 0 to denote the "mask" token.

**Forward process.** For a given $x_0 \sim p_{\text{data}}$ and a noise level $t \in [0, 1]$, the forward process $x_t \sim q_{t|0}(\cdot \mid x_0)$ is a coordinate-independent masking process via $q_{t|0}(x_t|x_0) = \prod_{i=0}^{L-1} q_{t|0}(x_t^i|x_0^i)$, where $q_{t|0}(x_t^i \mid x_0^i) = \text{Cat}(\alpha_t \mathbf{e}_{x_0^i} + (1-$

## MDM training

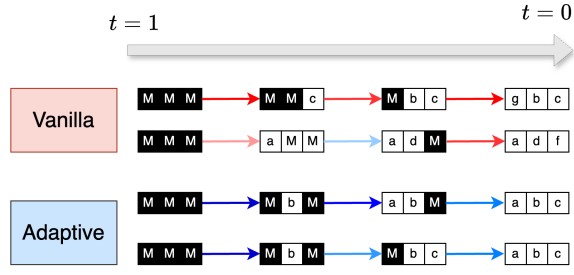

## MDM inferences (Vanilla vs. Adaptive)

*Figure 1.* (**Top**) MDM training can be seen as learning multiple masked prediction problems, where some are harder to learn, leading to performance imbalance (Section 3). (**Bottom**) During inference, adaptive MDM can avoid difficult problem instances, improving performance (Section 4).

$\alpha_t)\mathbf{e}_0)$ and $\alpha_t$ is the predefined noise schedule satisfying $\alpha_0 \approx 1, \alpha_1 \approx 0$ and $\mathbf{e}_{x_0^i} \in \mathbb{R}^{m+1}$ denotes a one-hot vector corresponding to the value of token $x_0^i$. $\text{Cat}(\pi)$ denotes the categorical distribution given by $\pi \in \Delta^m$. In other words, for each $i$-th coordinate, $x_t^i$ is masked to the mask token 0 with probability $1 - \alpha_t$ and unchanged otherwise.

**Reverse process.** The reverse process of the above forward process is denoted using $q_{s|t}(x_s|x_t, x_0)$ and is given by $q_{s|t}(x_s|x_t, x_0) = \prod_{i=0}^{L-1} q_{s|t}(x_s^i|x_t, x_0)$ for any $s < t$, where

$$q_{s|t}(x_s^i \mid x_t, x_0) = \begin{cases} \text{Cat}(\mathbf{e}_{x_t^i}) & x_t^i \neq 0 \\ \text{Cat}\left(\frac{1-\alpha_s}{1-\alpha_t}\mathbf{e}_m + \frac{\alpha_s - \alpha_t}{1-\alpha_t}\mathbf{e}_{x_0}\right) & x_t^i = 0. \end{cases}$$

The reverse transition probability $q_{s|t}(x_s^i|x_t, x_0)$ is approximated using $g_\theta(x_s^i|x_t) \triangleq q_{s|t}(x_s^i \mid x_t, x_0 \leftarrow p_\theta(x_t, t))$ where $p_\theta(x_t, t)$ is a denoising network trained to predict the marginal on $x_0$ via an ELBO-based loss. To be precise, $q_{s|t}(x_s^i \mid x_t, x_0 \leftarrow p_\theta(x_t, t))$ indicates the conditional probability where $p_\theta(x_t, t)$ is placed in the position of $x_0$

within $q_{s|t}(x_s^i \mid x_t, x_0)$.

$$\mathcal{L}_\theta = \int_0^1 \frac{\alpha_t'}{1-\alpha_t} \underset{\substack{x_0 \sim p_{\text{data}} \\ x_t \sim q_{t|0}(\cdot|x_0)}}{\mathbb{E}} \sum_{i:x_t^i=0} -\log p_\theta(x_0^i|x_t, t)dt.$$

Here, $\alpha_t' = \frac{d\alpha_t}{dt}$ and $\delta_{x_t,0}$ is the indicator function; the summation is computed over coordinates $i$ s.t. $x_t^i = 0$. In practice, a time-embedding-free architecture for the denoising network, i.e., $p_\theta(x_t, t) = p_\theta(x_t)$, is usually employed as $x_t$ implicitly contains information about $t$ via the number of masked tokens.

The reverse sampling process starts from the fully masked sentence $x_1 = (0, \ldots, 0)$. At a given noise level $t \in (0, 1]$, suppose we have a partially masked sequence $x_t$. For predetermined noise level $s < t$, we sample $x_s \sim g_\theta(\cdot|x_t)$. This process is repeated recursively from $t = 1$ to $t = 0$.

## 2.1. Reformulating the training and inference of MDMs

In this section, we first discuss vanilla order-agnostic training of MDMs and compare it with "left-to-right" order training of autoregressive models in Section 2.1.1. Then, we reformulate vanilla MDM inference in Section 2.1.2 to set the stage for the upcoming discussion.

### 2.1.1. ORDER-AGNOSTIC TRAINING OF MDMs

Recent works (Zheng et al., 2024; Ou et al., 2024) have observed that the learning problem of MDM is equivalent to a masked language model. Building upon their analysis, we reformulate the loss $\mathcal{L}_\theta$ to show that $\mathcal{L}_\theta$ is a linear combination of the loss for all possible infilling masks. We first define $x_0[M]$ as a masked sequence, obtained from original sequence $x_0$ where indices in the mask set $M$ (regarded as a subset of $[L] \triangleq \{1, 2, \ldots, L\}$) are replaced with mask token 0.

**Proposition 2.1.** *Assume $\alpha_0 = 1$, $\alpha_1 = 0$ and denoising network $p_\theta$ is time-embedding free. Then $\mathcal{L}_\theta \leq -\mathbb{E}_{x_0 \sim p_{\text{data}}}[\log p_\theta(x_0)]$ and*

$$\mathcal{L}_\theta = -\sum_{M \subseteq [L], i \in M} \frac{1}{|M|} \frac{1}{\binom{L}{|M|}} \underset{x_0 \sim p_{\text{data}}}{\mathbb{E}} [\log p_\theta(x_0^i|x_0[M])],$$
(1)

*where $|M|$ is the size of the set $M$ and $p_\theta(x_i \mid x_0[M])$ indicates the conditional probability of the $i$-th coordinate from $p_\theta(x_t)$.*

The proof of the above proposition is given in Appendix E. As the MDM loss is a linear combination of the loss for all possible infilling mask $M$, the minimizer of the loss $\mathcal{L}_\theta$ learns to solve *every* masking problem. In other words, the optimal predictor $p_\theta$ is the posterior marginal of the $i$-th token, conditioned on $x_0[M]$ for all masks $M$. The training objective of MDM aims to predict $x_0$ from $x_0[M]$ across all

possible masks. Hence, we will refer to the MDM training as *order-agnostic* training.

On the other hand, Autoregressive Models (ARMs) learn to solve a smaller set of infilling problems ($L$ infilling problems in ARMs as opposed to $\exp(L)$ infilling problems in MDM) by predicting $i^{\text{th}}$ token $x^i$ given all previous tokens $x^0, \ldots, x^{i-1}$. This prediction problem is equivalent to predicting $x^i$ by masking at positions $\{i, \ldots, L-1\}$. Therefore, we can write it as

$$\log p_\theta(x_0) = \sum_{i=0}^{L-1} \log p_\theta(x_0^i|x_0[\{i, \ldots, L-1\}]). \quad (2)$$

ARMs are trained to predict tokens sequentially from left to right in all sequences. We refer to this as left-to-right training. In general, one can also consider predicting tokens sequentially under some *fixed, known* permutation of the sequence; we refer to this as *order-aware training*.

### 2.1.2. ORDER-AGNOSTIC INFERENCE OF MDMs

The MDM inference can be decomposed into two steps: (a) randomly selecting a set of positions to unmask and (b) assigning token values to each position via the denoising network $p_\theta$. More precisely, we can reformulate the reverse process $x_s \sim g_\theta(\cdot|x_t)$ as follows.

---

**Vanilla MDM inference**

(a) Sample a set of masked tokens $\mathcal{S} \subseteq \{i \mid x_t^i = 0\}$, $\mathbb{P}(i \in \mathcal{S}) = \frac{\alpha_s - \alpha_t}{1 - \alpha_t}$.

(b) For each $i \in \mathcal{S}$, sample $x_s^i \sim p_\theta(x^i|x_t)$.

---

Therefore, the inference in MDM is implemented by randomly selecting $S$ and then filling each token value according to the posterior probability $p_\theta(x_s^i|x_t)$.

## 3. MDMs train on hard problems

In this section, we theoretically and empirically demonstrate that a large portion of masking subproblems $p_\theta(x_0^i \mid x_0[M])$ can be difficult to learn. For intuition, consider solving a masked prediction problem $p_\theta(x^i \mid x_0[M])$ on text data like masking an arbitrary sentence in the middle of a document and predicting the correct word for a specific position in that sentence. It is reasonable that this task should be more complex, even for humans, than left-to-right prediction, and in this section, we place this intuition on a rigorous footing.

In Section 3.1, we show several examples of simple, non-pathological distributions for which: (1) the masking problems encountered during order-*aware* training are computationally tractable, yet (2) many of the ones en-

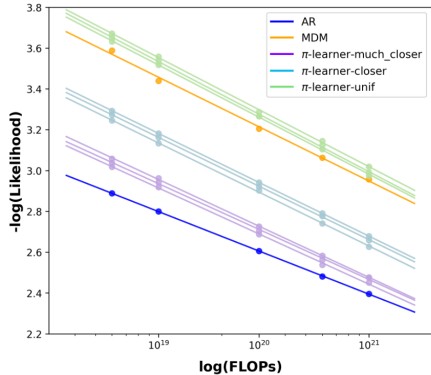

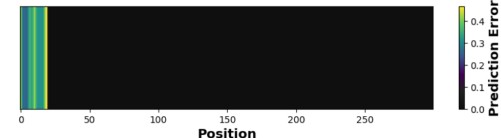

*Figure 2.* **Left: MDMs train on hard problems (Section 3.2)**. x-axis and y-axis correspond to $\log(\text{FLOPs})$ and $-\log p_\theta(x)$, respectively. MDM (Blue) is worse than ARM (Orange) in likelihood modeling. Most masking problems (Other lines) that MDM is trained on are harder than those encountered by ARM, as indicated by small log-likelihoods. **Right: Task error imbalance (Section 3.3)**. MDM's performance varies across different tasks. For text data (top right), this is indicated by validation loss. For L&O-NAE-SAT (bottom right), MDM performs well on the masking problems for observation positions (light region) but struggles with latent positions (dark region).

countered during order-agnostic training are computationally intractable. In Section 3.2, we empirically show that text data also exhibits this gap between the computational complexity of order-aware and order-agnostic training. In Section 3.3, we reveal that this discrepancy in computational complexity manifests empirically in **performance imbalance across tasks**: as predicted by the theory, MDMs trained on data from such distributions exhibits small errors on easy subproblems but suffers from large errors on harder ones.

### 3.1. Benign distributions with hard masking problems

We now describe a simple model of data under which we explore the computational complexity of masking problems.

**Definition 3.1.** A *latents-and-observations (L&O) distribution* is a data distribution $p_{\text{data}}$ over sequence of length $L$ with alphabet size $m$ (precisely, $p_{\text{data}}$ is over $\{0,\ldots,m\}^L$) is specified by a permutation $\pi$ over indices $\{1,2,\ldots,L\}$, number of latent tokens $N$, number of observation tokens $P$ such that $N + P = L$, prior distribution $p_{\text{prior}}$ of latent variables over $\{1,\ldots,m\}$ and efficiently learnable *observation functions* $\mathcal{O}_1,\ldots,\mathcal{O}_P : \{1,\ldots,m\}^N \to \Delta(\{0,\ldots,m\})$,[1]

- (**Latent tokens**) For $i = 1,\ldots,N$, sample $x^{\pi(i)}$ independently from the prior distribution $p_{\text{prior}}$ of the latents.

- (**Observation tokens**) For $j = 1,\ldots,P$, sample $x^{\pi(N+j)}$ independently from $\mathcal{O}_j(x^{\pi(1)},\ldots,x^{\pi(N)})$.

L&O distributions contain two types of tokens: (1) *latent tokens* and (2) *bservation tokens*. Intuitively, latent tokens are

---

[1] Here *efficiently learnable* is in the standard PAC sense: given polynomially many examples of the form $(z,y)$ where $z \sim \pi^n$ and $y \sim \mathcal{O}_j(z)$, there is an efficient algorithm that can w.h.p. learn to approximate $\mathcal{O}_j$ in expectation over $\pi^n$.

tokens in the sequence, indexed by $\pi(1),\pi(2),\ldots,\pi(N)$ that serve as "seeds" that provide randomness in the sequence; the remaining tokens, called observation tokens (indexed by $\pi(N+1),\pi(N+2),\ldots,\pi(N+P)$), are determined as (possibly randomized) functions of the latent tokens via $\mathcal{O}_1,\ldots,\mathcal{O}_P$.

Note that by design, order-aware training, e.g. by permuting the sequence so that $\pi$ becomes the identity permutation and then performing autoregressive training, is computationally tractable: predicting $x^{\pi(i)}$ given $x^{\pi(1)},\ldots,x^{\pi(i-1)}$ is trivial when $i \leq N$ as the tokens are independent, and computationally tractable when $i > N$ because $x^{\pi(i)}$ only depends on $x^{\pi(1)},\ldots,x^{\pi(N)}$ and is efficiently learnable by assumption. In contrast, below we will show examples where if one performs order-agnostic training *à la* MDMs, one will run into hard masking problems with high probability.

First note that if the observations $(\mathcal{O}_1,\ldots,\mathcal{O}_P)$ are given by a cryptographic hash function, then the masking problem of predicting $(x^{\pi(1)},\ldots,x^{\pi(L)})$ given $(x^{\pi(N+1)},\ldots,x^{\pi(N+P)})$ is computationally intractable by design because it requires inverting the hash function. While this is a well-known folklore observation regarding the role of token ordering in language modeling, it is not entirely satisfying because this construction is worst-case in nature – in real-world data, one rarely trains on sequences given by cryptographic hash functions. Furthermore, it only establishes hardness for a specific masking pattern which need not be encountered in the course of running the reverse process.

We provide several simple instances of L&O distributions that address these issues: instead of leveraging delicate cryptographic constructions, they are *average-case* in nature and furthermore we can establish hardness for *typical* masking

problems encountered along the reverse process.

In all these examples, the hardness results we establish hold even if the algorithm knows all of the parameters of $p_{\text{data}}$ as well as the observation functions $\mathcal{O}_1, \ldots, \mathcal{O}_P$. Due to space constraints, here we focus on the following example, deferring two others to Apps. B.1 and B.2.

**Example 3.2** (Sparse predicate observations). *Consider the following class of L&O distributions. Given arity $k \geq 2$, fix a* predicate *function $g : \{1, \ldots, m\}^k \to \{0, 1\}$. Consider the set of all ordered subsets of $\{1, 2, \ldots, N\}$ of size $k$ and set the total number of observation latents $P$ equal to the size of this set (hence $P = N!/(N - k)! = N(N-1)\cdots(N-k+1)$). To sample a new sequence, we first sample latent tokens $x^{\pi(1)}, \ldots, x^{\pi(N)}$ from the prior distribution $p_{prior}$ and an observation latent corresponding to a $k$-sized subset $S$ is given by $g(\{x^{\pi(i)}\}_{i \in S})$. In other words, each observation latent corresponds to a $k$-sized subset $S$ of $\{1, 2, \ldots, N\}$ and the corresponding observation function $\mathcal{O}_S(x^{\pi(1)}, \ldots, x^{\pi(N)})$ is given by $g(\{x^{\pi(i)}\}_{i \in S})$.*

**Proposition 3.3.** *Let $x$ be a sample from an L&O distribution $p_{\text{data}}$ with sparse predicate observations as defined in Example 3.2, with arity $k$ and predicate $g$ satisfying Assumption B.11, and let $\gamma$ be the probability that $g$ is satisfied by a random assignment from $\{1, \ldots, m\}^k$. Let $D_{\text{KS}}$ and $D_{\text{cond}}$ be some constants associated with the predicate function $g$ (see Definition B.12). Suppose each token in $x$ is independently masked with probability $\alpha$, and $M$ is the set of indices for the masked tokens. If $1 - \gamma^{-1}D_{\text{KS}}/kN^{k-1} \leq \alpha \leq 1 - \gamma^{-1}D_{\text{cond}}/kN^{k-1}$, then under the 1RSB cavity prediction (see Conjecture B.13), with probability $\Omega_k(1)$ over the randomness of the masking, no polynomial-time algorithm can solve the resulting subproblem of predicting any of the masked tokens among $x^{\pi(1)}, \ldots, x^{\pi(N)}$ given $x[M]$.*

The complete proof of the proposition is given in Appendix B.4. We also provide a proof outline in Appendix B.3 for a comprehensive understanding.

### 3.2. Empirical evidence of hardness via likelihoods

Recent studies (Nie et al., 2024; Zheng et al., 2024) have shown that masked diffusion models (MDMs) underperform compared to autoregressive models (ARMs) on natural text data. In this section, we provide evidence that this performance gap is primarily due to the order-agnostic training of MDMs. Since natural text follows a left-to-right token order, we demonstrate that as training deviates from this order, model performance gradually deteriorates.

To understand the importance of the order during the training, we use the following setting: Given a permutation $\pi$ of indices $\{0, 1, \ldots, L - 1\}$, define a $\pi$-*learner* to be a

likelihood model $\log p_\theta(x_0)$ given as follows:

$$\log p_\theta(x_0) = \sum_{i=0}^{L-1} \log p_\theta\left(x_0^{\pi(i)} \Big| x_0[\pi\{i, \ldots, L-1\}]\right) \quad (3)$$

In other words, the $\pi$-learner predicts the token at position $\pi(i)$ given the clean tokens $x_0^{\pi(0)}, \ldots, x_0^{\pi(i-1)}$ and masked tokens $x_0^{\pi(i)}, \ldots, x_0^{\pi(L-1)}$. If $\pi$ is the identity permutation, this reduces to the standard (left-to-right) autoregressive model. Note that the MDM loss encodes a $\pi$-learner for every permutation $\pi$ because the MDM loss (1) is equivalent to the average loss of those $\pi$-learners over $\pi$ sampled from $\text{Unif}(\mathbb{S}_L)$:

$$\mathcal{L}_\theta = -\mathbb{E}_{\pi, x_0 \sim p_{\text{data}}}\left[\sum_{i=0}^{L-1} \log p_\theta\left(x_0^{\pi(i)} \Big| x_0[\pi\{i, \ldots, L-1\}]\right)\right],$$

where $\mathbb{S}_L$ denotes the set of all permutations over $\{0, 1, \ldots, L - 1\}$. The proof of the above equivalence is given in Appendix E. Therefore, by measuring the 'hardness' of each $\pi$-learner, we can probe differences in hardness between arbitrary masking problems and left-to-right masking problems.

**Experimental setup.** We use the Slimpajama dataset (Soboleva et al., 2023) to evaluate the performance of training in different orders. To train a $\pi$-learner, we employ a transformer with causal attention and use permuted data $\pi(x_0)$ as input. By varying $\pi$ while maintaining all other training configurations (e.g., model, optimization), we can use the resulting likelihood (computed using Equation (3)) as a metric to capture the hardness of subproblems solved by the $\pi$-learner.

In our experiments, the sequence length $L$ is approximately $10^3$, so repeating the above for each $\pi$ is infeasible. Instead, we sample $\pi \sim \text{Unif}(\mathbb{S}_L)$ and examine the scaling law of the $\pi$-learner's likelihood. We leverage the codebase from (Nie et al., 2024), where the baseline scaling laws of MDM and ARM were introduced. Moreover, given that RoPE has an inductive bias towards left-to-right ordering, we employ a learnable positional embedding layer for all experiments to correct this. Consequently, we also re-run the baseline results, where RoPE was employed. To investigate how the distance between $\pi$ and the identity permutation affects the scaling law, we sample $\pi$ from other distributions interpolating between $\text{Unif}(\mathbb{S}_L)$ and the point mass at the identical permutation. Further experimental details are provided in Appendix C.1.

**Results.** As shown in Fig. 2, the scaling law for a $\pi$-learner with uniformly random $\pi$ is worse than that of an ARM. This elucidates the inherent hardness of masking problems $p_\theta(x_i \mid x_0[M])$ beyond left-to-right prediction and also explains why MDM, which is trained simultaneously on

all $\pi \in \mathbb{S}_L$, is worse than ARM in likelihood modeling. Additionally, as $\pi$ gets closer to the identity permutation, the scaling laws also get closer to ARM ($\pi$-learner-closer and $\pi$-learner-much-closer in Fig. 2). This also supports the common belief that ARM is a good fit for text data as it inherently follows a *left-to-right* ordering.

That said, it should also be noted that even though MDMs are trained on exponentially more masking problems than ARM ($\Theta(L2^L)$ versus $L$), its performance is not significantly worse than $\pi$-learners. We attribute this to the *blessing of task diversity*; multi-task training can benefit both the optimization dynamics (Kim et al., 2024) and validation performance (Tripuraneni et al., 2021; Maurer et al., 2016; Ruder, 2017) due to positive transfers across tasks.

### 3.3. Error is imbalanced across masking problems

In previous sections, we have demonstrated that the hardness of different masking problems $p_\theta(x^i \mid x_0[M])$ can vary significantly, potentially hindering the MDM's learning. In this section, we provide empirical evidence that the MDM's final performance exhibits a similar imbalance across subproblems. Details are provided in App. C.2.

**L&O-NAE-SAT.** Consider an L&O distribution with $\pi$ given by the identity permutation and where each observation $\mathcal{O}_j$ is deterministically given by $\mathrm{NAE}(x_{i_1}, x_{i_2}, x_{i_3}) \triangleq 1 - \mathbf{1}[x_{i_1} = x_{i_2} = x_{i_3}]$ for some randomly chosen (prefixed) triples $(i_1, i_2, i_3) \in [N]$. For an MDM trained on this distribution, we measure the error it achieves on each task $\log p_\theta(x_0|x_0[M])$ via $\mathbb{E}_{x_0} \big\| \log p_\theta(x_0|x_0[M]) - \log p_{\mathrm{data}}(x_0|x_0[M]) \big\|^2$, where $p_{\mathrm{data}}(x_0|x_0[M])$ denotes the Bayes-optimal predictor. Technically, we do not have access to this, so instead we train another MDM for a much larger number of iterations and use this as a proxy. Fig. 2 reveals that prediction tasks for latent positions (light region) exhibit larger errors compared to those for observation positions (dark region).

**Text.** Here we revisit the text experiment from Section 3.2. Since we do not have access to the Bayes-optimal predictor, we use the metric $\mathbb{E}_{x_0 \sim p_{\mathrm{data}}} \left[ \sum_{i=0}^{L-1} \log p_\theta \left( x_0^{\pi(i)} \middle| x_0[\pi\{i, \dots, L-1\}] \right) \right]$. This captures the accumulation of error across subproblems $p_\theta \left( x_0^{\pi(i)} \middle| x_0[\pi\{i, \dots, L-1\}] \right)$, since $p_\theta(x_0|x_0[M]) = p_{\mathrm{data}}(x_0|x_0[M])$ minimizes this metric. Fig. 2 shows a clear gap between different subproblems.

The theoretical and empirical evidence demonstrates that MDMs perform better in estimating $p_\theta(x_0|x_0[M])$ for some subproblems $M$ than for others. We therefore want to avoid encountering hard subproblems $M$ at inference time. In the next section, we show that while vanilla MDM inference can run into such subproblems, simple modifications at the inference stage can effectively circumvent these issues, resulting in dramatic, *training-free* performance improvements.

## 4. MDMs can plan around hard problems

We previously argued that due to the complex nature of masking subproblems, MDM must perform poorly on certain ones $p_\theta(x^i|x_t)$. Therefore, during vanilla MDM inference, MDM inevitably encounters such difficult subproblems at Step (b). While this might suggest that we need to fundamentally revisit how MDMs are trained, in this section we show that, surprisingly, simple modifications at the inference stage—*without any further training*—can sidestep these issues and lead to significant performance improvements.

**MDM offers multiple sampling paths.** The vanilla MDM inference (Algorithm 1) aim to align the intermediate distributions with the forward process, as used in continuous diffusion. However, unlike continuous diffusion, the reverse process of MDM allows multiple valid sampling paths (different orders of unmasking the tokens) that match the starting distribution of the forward process of MDM.

We first show that when we have an ideal MDM that perfectly solves all masking problems, i.e., $p_\theta(x_0^i|x_0[M]) = p_{\mathrm{data}}(x_0^i|x_0[M])$, then using any sampling path (unmasking the tokens in any order) results in the same distribution. Consider the following sampler: For every step, $S$ is a set with one index selected agnostically (without following any distribution). For any clean sample $x_0$ generated by this sampler, note that $p_\theta(x_0) = \prod_{i=0}^{L-1} p_\theta \left( x_0^{\pi(i)} \middle| x_0[\pi\{i, \dots, L-1\}] \right)$ by chain rule, and this is equal to $\prod_{i=0}^{L-1} p_{\mathrm{data}} \left( x_0^{\pi(i)} \middle| x_0[\pi\{i, \dots, L-1\}] \right) = p_{\mathrm{data}}(x_0)$. Therefore, other choices of $S$, not necessarily following Algorithm 1, still capture the true likelihood.

In practice, unlike this ideal case, MDM does not perform equally well on all subproblems, as shown in Section 3.3. Consequently, different sampling paths result in varying likelihood modeling abilities. Motivated by this observation, we consider *adaptive inference for MDMs*:

---

**Adaptive MDM inference**

(a) Sample a set of masked tokens $\mathcal{S} = \mathcal{F}(\theta, x_t) \subseteq \{i \mid x_t^i = 0\}$.

(b) For each $i \in \mathcal{S}$, sample $x_s^i \sim p_\theta(x^i|x_t)$.

---

Instead of selecting $S$ randomly, adaptive MDM inference leverages an oracle $\mathcal{F}(\theta, x_t)$ to select $S$ strategically to

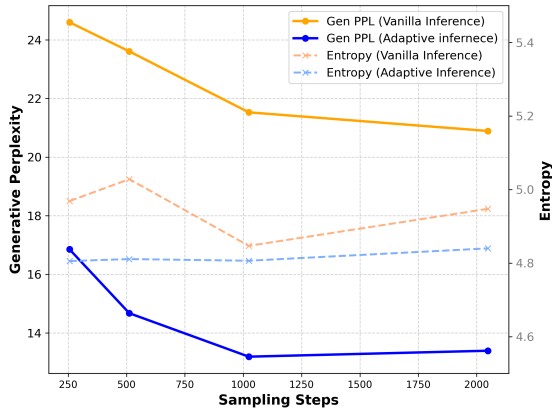

*Figure 3.* **Generative Perplexity.** We compare the resulting generative perplexity (GenPPL) of adaptive vs. vanilla MDM inference. We employ a pretrained 170M MDM and LLaMA-7B (Touvron et al., 2023) as inference and evaluation, respectively. Adaptive MDM inference (Blue) leads to a substantial reduction in generative perplexity, while maintaining the entropy.

avoid hard masking problems. This naturally raises the question of how to design an effective oracle $\mathcal{F}$.

In the following sections, we demonstrate that adaptive MDM inference with careful choices of $\mathcal{F}$ enhance MDM's likelihood matching ability. In other words, a pretrained MDM, even if it performs poorly on certain hard subproblems, **still contains sufficient information to avoid them** when paired with an effective oracle $\mathcal{F}$.

### 4.1. Effective design of ordering oracle

We introduce two different oracles, Top-$K$ and Top-$K$ probability margin. Intuitively, both strategies are based on the idea that $S$ should be selected based on how "certain" the model is about each position. We caution that these strategies should not be confused with notions like nucleus sampling in ARMs (Holtzman et al., 2019); the oracles we describe are for selecting the *position* of the next token to decode, rather than the *value*, and thus are only meaningful in the context of MDMs.

*Table 1.* **L&O-NAE-SAT**. Adaptive MDM inference achieves better likelihood matching than vanilla MDM inference. Note that naive guessing leads to 75% accuracy, indicating that vanilla inference performs similarly or worse than naive guessing.

| $(N, P)$ | Vanilla inference | Adaptive inference |
|---|---|---|
| $(25, 275)$ | 78.06% | 93.76% |
| $(30, 270)$ | 75.70% | 93.54% |
| $(40, 260)$ | 74.60% | 92.21% |
| $(50, 250)$ | 67.94% | 90.01% |
| $(100, 200)$ | 62.84% | 88.91% |

*Table 2.* Comparison of accuracy for solving the Sudoku puzzle.

| Method | # Param | Accuracy |
|---|---|---|
| ARM (w/o ordering) | 42M | 9.73% |
| ARM (with ordering) | | 87.18% |
| MDM (vanilla) | | 6.88% |
| MDM (Top-$K$ probability) | 6M | 18.51% |
| MDM (Top-$K$ prob. margin) | | 89.49% |

**Top-$K$ probability (Zheng et al., 2023).** Suppose we want to unmask $K$ positions at time step $t$, i.e., select $|S| = K$. In the Top-$K$ strategy, the uncertainty of a position is estimated by the maximum probability assigned to any value in the vocabulary. More precisely, the certainty at position $i$ is $\max_{j \in \{0,...,m-1\}} p_\theta(x^i = j|x_t)$ and $\mathcal{F}(\theta, x_t) = \text{Top } K \left( \max p_\theta(x^i|x_t) \right)$.

Top-$K$ strategy is a good proxy for many tasks and works well in practice (Zheng et al., 2023; Ye et al., 2024; Wang et al., 2024). However, this approach can often provide misleading estimates of uncertainty. Consider when an MDM is confused between two token values, thus assigning them almost equal but high probabilities. In this case, Top-$K$ strategy may still choose to unmask this position, despite its uncertainty. To mitigate this issue, we propose the following alternative strategy.

**Top-$K$ probability margin.** In this strategy, the uncertainty of a position is instead estimated using the absolute difference between the two most probable values at position $i$. More precisely, if $j_1$ and $j_2$ are the two most probable values in vocabulary according to $p_\theta(x^i|x_t)$ in position $i$, the certainty in the position is given by $|p_\theta(x^i = j_1|x_t) - p_\theta(x^i = j_2|x_t)|$ and $\mathcal{F}(\theta, x_t) = \text{Top } K \left( |p_\theta(x^i = j_1|x_t) - p_\theta(x^i = j_2|x_t)| \right)$. When multiple values have similar probabilities at a position, Top-$K$ probability margin will provide a better estimate of the uncertainty of a position, and when there is a single best choice of value then Top-$K$ and Top-$K$ probability margin work similarly.

### 4.2. Adaptive MDM inference

In this section, we experimentally validate that adaptive MDM inference helps MDMs avoid hard subproblems, leading to better likelihood matching. We first show our results on L&O-NAE-SAT and text data, before turning to our primary application to logic puzzles.

**L&O-NAE-SAT and text data.** For the L&O-NAE-SAT distribution defined in Section 3.3, we evaluate the effectiveness of adaptive inference by measuring the accuracy in predicting the observation tokens. Table 1 in the appendix

reveals a clear improvement over vanilla inference. For the text dataset, we evaluate using the standard metric of *generative perplexity*, by which likelihood is measured by a large language model. We also compute the entropy of the generated samples to ensure both inference strategies exhibit similar levels of diversity. As shown in Fig. 3, we observe a substantial decrease in generative perplexity using adaptive inference. We defer further experimental details to Appendix D.1.

**Logic puzzles.** We consider two different types of logic puzzles: Sudoku and Zebra (Einstein) puzzles. Intuitively, for Sudoku, some empty (masked) cells are significantly easier to predict than others and we want to choose the cells that are easier to predict during the inference. We evaluate the effectiveness of adaptive MDM inference over vanilla MDM inference in selecting such cells.[2]

To measure the performance of an inference method, we use the percentage of correctly solved puzzles. For both puzzles, we use train and test datasets from (Shah et al., 2024). For the Sudoku puzzle (Table 2) we observe that adaptive MDM inference, in particular Top-$K$ probability margin, obtains substantially higher accuracy (89.49%) compared to vanilla MDM inference (6.88%). Additionally, Top-$K$ probability margin obtains higher accuracy (89.49%) than Top-$K$ (18.51%). As mentioned in Section 4.1, this is because Top-$K$ probability margin more reliably estimates uncertainty when multiple competing values are close in probability at a given position, as is often the case in Sudoku. For the Zebra puzzle, as shown in Table 3, we observe a consistent result: Top-$K$ (98.5%) and Top-$K$ probability margin (98.3%) outperform vanilla MDM inference (76.9%).

*Table 3.* Comparison of accuracy for solving the Zebra puzzle.

| Method | # Param | Accuracy |
|---|---|---|
| ARM (w/o ordering) | 42M | 80.31 % |
| ARM (with ordering) | | 91.17 % |
| MDM (vanilla) | 19M | 76.9 % |
| MDM (Top-$K$ probability) | | 98.5 % |
| MDM (Top-$K$ prob. margin) | | 98.3 % |

### 4.3. Eliciting sequence-dependent reasoning paths using adaptive MDM inference in logic puzzles

In this section, we study the effectiveness of adaptive MDM inference in finding the right reasoning/generation order for tasks where every sequence has a different "natural" order.

---

[2]A prior work (Ye et al., 2024) reported that a 6M MDM with Top-$K$ inference achieves 100% accuracy on Sudoku. Given that a 6M MDM with Top-$K$ only achieves 18.51% on our dataset (Table 2), this suggests that the Sudoku dataset in (Ye et al., 2024) is significantly easier than ours.

To do so, we will compare the performance of adaptive MDM inference to that of ARM on Sudoku and Zebra puzzles. For these puzzles, the natural order of generation is not only different from left-to-right, but it is also sequence-dependent. For such tasks, prior works have shown that ARMs struggle if the information about the order is not provided during the training (Shah et al., 2024; Lehnert et al., 2024). Therefore, to obtain a strong baseline, we not only consider an ARM trained without the order information but also consider an ARM trained with the order information for each sequence in the training data. Note that the latter is a much stronger baseline than the former as one can hope to teach the model to figure out the correct order by some form of supervised teacher forcing (as performed in Shah et al. (2024); Lehnert et al. (2024)), eliminating the issue of finding the right order in an unsupervised manner.

We compare ARMs and MDMs for Sudoku in Table 2 and Zebra puzzles in Table 3. We observe that for both, Top-$K$ probability margin-based adaptive MDM inference not only outperforms the ARM trained without ordering information, but it *even outperforms the ARM trained with ordering information*! This shows that the *unsupervised* way of finding the correct order and solving such logic puzzles using adaptive MDM inference outperforms the *supervised* way of finding the correct order and solving such puzzles using an ARM, and is significantly less computationally intensive.

### 4.4. Adaptive MDM inference on text benchmarks

To examine the effect of different inference strategies on text benchmarks, we adapted LLaDA, the 8B MDM model from (Nie et al., 2025). We compare three inference strategies: Vanilla, Top-$K$ probability, and Top-$K$ probability margin . The results are presented in Table 4.

We see that both adaptive MDM inference strategies, Top-$K$ probability and Top-$K$ probability margin , consistently outperform vanilla MDM inference. Notably, Top-$K$ probability margin demonstrates a clear advantage over Top-$K$ in challenging tasks like HumanEval-Multiline, HumanEval-Split Line, and Math. This is because Top-k Prob. Margin provides a more reliable estimate of uncertainty when multiple tokens have similar probabilities, a frequent occurrence in these difficult tasks. These results further underscore the potential for developing new, sophisticated adaptive inference strategies for various tasks.

### 4.5. Easy to hard generalization

In the previous section we showed that when the training and inference sequences come from the same distribution, order-agnostic training of MDMs combined with adaptive inference can perform very well on logic puzzles. To evaluate if the model has learned the correct way of solving the puzzles and test the robustness of adaptive inference, we

*Table 4.* Performance of different inference strategies for LLaDa 8B model on coding and math tasks.

| Method | HumanEval-Single | HumanEval-Multi | HumanEval-Split | Math | MMLU | ROCStories |
|---|---|---|---|---|---|---|
| Vanilla | 31.8% | 16.5% | 14.2% | 28.5% | 33.2% | 21.23% |
| Top-k | 32.9% | 20.8% | 18.4% | 31.3% | **36.5%** | 21.10% |
| Top-k Margin | **33.5%** | **25.4%** | **22.3%** | **34.3%** | 35.4% | **21.41%** |

*Table 5.* Comparison of accuracy for solving the hard Sudokus.

| Method | #Param | Accuracy |
|---|---|---|
| ARM (with ordering) | 42M | 32.57 % |
| MDM (random) | | 3.62 % |
| MDM (Top-$K$ probability) | 6M | 9.44 % |
| MDM (Top-$K$ prob. margin) | | 49.88 % |

also test the MDMs on harder puzzles than the ones from training, for Sudoku.

We keep the training dataset the same as proposed in Shah et al. (2024). Shah et al. (2024) created this dataset from Radcliffe (2020) by selecting the puzzles that can be solved using 7 fixed strategies and do not require backtracking-based search. We use the remaining puzzles in Radcliffe (2020) as our hard dataset. Hence, these puzzles all use a strategy not seen during training and/or backtracking to obtain the correct solution.

We measure the accuracy of MDMs and ARMs on the hard test set and present the results in Table 5. We see that the Top-$K$ probability margin-based adaptive MDM inference strategy (49.88%) again significantly outperforms ARMs trained with order information (32.57%). In particular, although the accuracy drops for both methods due to the more challenging test set, MDMs with adaptive inference appear to be more robust to this distribution shift than ARMs. We believe this is due to the fact that MDMs try to solve a significantly higher number of infilling problems than ARMs ($\exp(L)$ compared to $L$) and therefore are able to extract knowledge about the problem more efficiently than ARMs.

## 5. Conclusion

In this work, we examined the impact of token ordering on training and inference in MDMs. We provided theoretical and experimental evidence that MDMs train on hard masking problems. We also demonstrated that adaptive inference strategies can be used to sidestep these hard problems. For logic puzzles, we find that this leads to dramatic improvements in performance not just over vanilla MDMs, but even over ARMs trained with teacher forcing to learn the right order of decoding.

An important direction for future work is to explore settings beyond logic puzzles where adaptive inference can help MDMs match or surpass ARMs. For these, it may be crucial to go beyond the relatively simple adaptive strategies like Top-$K$ and Top-$K$ probability margin considered here.

**Acknowledgements.** JK thanks Kiwhan Song for discussions about MDM training. KS and VK are supported by the NSF AI Institute for Foundations of Machine Learning (IFML). KS thanks Nishanth Dikkala for the initial discussions about the project. SC is supported by the Harvard Dean's Competitive Fund for Promising Scholarship and thanks Brice Huang and Sidhanth Mohanty for enlightening discussions about computational-statistical tradeoffs for planted CSPs.

## Impact statement

This paper advances the understanding of discrete diffusion models, contributing to the broader field of Machine Learning. There are many potential societal consequences of our work, none of which we feel must be specifically highlighted here.

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

# A. Related works

**Discrete diffusion models.** (Continuous) diffusion models were originally built on continuous-space Markov chains with Gaussian transition kernels (Sohl-Dickstein et al., 2015; Ho et al., 2020). This was later extended to continuous time through the theory of stochastic differential equations (Song et al., 2021). In a similar vein, discrete diffusion models have emerged from discrete-space Markov chains (Hoogeboom et al., 2021b). Specifically, (Austin et al., 2021) introduced D3PM with various types of transition matrices. Later, Lou et al. (2024) proposed SEDD, incorporating a theoretically and practically robust score-entropy objective. Additionally, Varma et al. (2024); Liu et al. (2024b) introduced novel modeling strategies that classify tokens in a noisy sequence as either signal (coming from clean data) or noise (arising from the forward process). In particular, Liu et al. (2024b) uses this to give a *planner* that adaptively determines which tokens to denoise. While this is similar in spirit to our general discussion about devising adaptive inference strategies, we emphasize that their approach is specific to discrete diffusions for which the forward process *scrambles* the token values, rather than masking them.

**Masked diffusion models.** Meanwhile, the absorbing transition kernel has gained popularity as a common choice due to its better performance than other kernels. Building on this, Sahoo et al. (2025); Shi et al. (2024) aligned its framework with continuous diffusion, resulting in a simple and principled training recipe, referring to it as *Masked Diffusion Model*. Subsequent studies have explored various aspects of MDM. Gong et al. (2024) efficiently trained MDM via adaptation from autoregressive models, scaling MDM up to 7B parameters. Zheng et al. (2024) interpreted MDMs as order-agnostic learners and proposed a first-hitting sampler based on this insight. Ye et al. (2024); Gong et al. (2024) demonstrated that MDM outperforms autoregressive models in reasoning and planning tasks, emphasizing its impact on downstream applications. Nie et al. (2024) examined the scaling laws of MDM, while Xu et al. (2024); Liu et al. (2024a) identified limitations in capturing coordinate dependencies when the number of sampling steps is small and proposed additional modeling strategies to address this issue. Schiff et al. (2024) studied conditional generation using MDM and Rector-Brooks et al. (2024) tackled the challenge of controlling generated data distributions through steering methodologies. Chen & Ying (2024) provided a theoretical analysis showing that sampling error is small given accurate score function estimation.

**Any-order reasoning.** Even though language tasks generally have a natural order of "left-to-right" token generation, in many tasks like planning, reasoning, and combinatorial optimization, the natural order of token generation can be quite different from "left-to-right". Even though prominent autoregressive-based language models achieve impressive performance on various tasks, many works (Golovneva et al., 2024; Chen et al., 2024; Kitouni et al., 2025) have shown that this performance is tied to the training order of the tasks and therefore can cause brittleness from it. For example, Chen et al. (2024) showed that simply permuting the premise order on math tasks causes a performance drop of 30%. The reason behind such brittleness regarding the ordering is the inherent "left-to-right" nature of the autoregressive models. Several works (Liao et al., 2020) have tried to address this issue in the autoregressive framework. In particular, (Papadopoulos et al., 2024) highlighted the significance of left-to-right ordering in natural language by comparing its likelihood to that of the reverse (right-to-left) ordering.

Recently, discrete diffusion models have emerged as a promising approach for discrete data apart from autoregressive models. Additionally, the order-agnostic training of discrete diffusion models opens up the multiple sampling paths during the inference but it also faces some challenges during the training therefore, they seem a promising approach to elicit any order reasoning. Zheng et al. (2023) proposed different ways of implementing an adaptive inference strategy for MDM but a *concrete understanding of why such an adaptive inference strategy is needed is still lacking*. In this work, we explore various aspects of vanilla MDM training and how adaptive MDM inference can mitigate the issues raised by vanilla MDM training and elicit any order reasoning.

We also want to mention the concurrent work by Peng et al. (2025) that proposes an alternative adaptive inference strategy by selecting $\mathcal{F}(\theta, x_t)$ based on the BERT model or the denoiser itself. In particular, Peng et al. (2025) uses the BERT model or the denoiser to obtain the uncertainty of a token and then uses Top-$K$ to decide the positions to unmask it. In contrast to their work, we disentangle the impact of token ordering on MDM training vs. MDM inference and provide a more complete understanding of the motivations for and benefits of adaptive inference. Additionally, our results indicate drawbacks to using Top-$K$ strategy as opposed to Top-$K$ margin in deciding which tokens to unmask when there are multiple values with high probabilities.

**Beyond autoregressive models.** Efforts to learn the natural language using non-autoregressive modeling began with BERT (Devlin et al., 2019). Non-causal approaches can take advantage of the understanding the text data representation.

(Chang et al., 2022) adopted a similar approach for learning image representations. Building on these intuitions, (Shih et al., 2022; Hoogeboom et al., 2021a) proposed any-order modeling, which allows a model to generate in any desired order. Shih et al. (2022) made the same observation that any-order models by default have to solve exponentially more masking problems than autoregressive models. However, whereas our work shows that learning in the face of this challenging task diversity can benefit the model at inference time, their work sought to alleviate complexity at training time by reducing the number of masking problems that need to be solved.

## B. Technical details from Section 3

**Notations.** Throughout this section, we use $x^i$ to denote the $i$-th coordinate of the vector $x$ and $z(j)$ to denote the $j$-th example. The $i$-th coordinate of the vector $z(j)$ is denoted by $z(j)^i$.

### B.1. Additional example: sparse parity observations

**Example B.1** (Noisy sparse parity observations). *Let $m = 2$, $k \in \mathbb{N}$, and $N^2 \log N \ll P \leq N^{0.49k}$. Fix noise rate $\eta > 0$ as well as strings $z(1), \ldots, z(P)$ sampled independently and uniformly at random from the set of $k$-sparse strings in $\{0, 1\}^N$. For each $j \in [P]$, define $\mathcal{O}_j(x)$ to be the distribution which places mass $1 - \eta$ on 1 (resp. 2) and mass $\eta$ on 2 (resp. 1) if $\sum_i x^i z(j)^i$ is odd (resp. even). Note that for $k = O(1)$, each of these observations is efficiently learnable by brute-force.*

Below we show that for a certain range of masking fractions, a constant fraction of the masking problems for the corresponding L&O distributions are computationally hard under the *Sparse Learning Parity with Noise* assumption (Alekhnovich, 2003). Formally we have:

**Proposition B.2.** *Let $0 < \alpha < 1$ be an arbitrary absolute constant, and let $\eta = 1/\mathrm{poly}(N)$ be sufficiently large. Let $x$ be a sample from a L&O distribution $p_{\mathrm{data}}$ with noisy parity observations as defined in Example B.1. Suppose each token is independently masked with probability $\alpha$, and $M$ is the set of indices for the masked tokens. If $1 - 1/N \leq \alpha \leq 1 - 1/2N$, then under the Sparse Learning Parity with Noise (SLPN) assumption (see Definition B.3), with constant probability over $M$, no polynomial-time algorithm can solve the resulting masking problem of predicting any of the masked tokens among $x^{\pi(1)}, \ldots, x^{\pi(N)}$ given $x[M]$.*

We note that it is important for us to take the observations to be *sparse* parities and to leverage the *Sparse* Learning Parity with Noise assumption. If instead we used *dense* parities and invoked the *standard* Learning Parity with Noise (LPN) assumption, we would still get the hardness of masking problems, but the observations themselves would be hard to learn, assuming LPN. This result is based on the following standard hardness assumption:

**Definition B.3** (Sparse Learning Parity with Noise). Given input dimension $N$, noise parameter $0 < \eta < 1/2$, and sample size $P$, an instance of the *Sparse Learning Parity with Noise (SLPN)* problem is generated as follows:

- Nature samples a random bitstring $x$ from $\{0, 1\}^N$

- We observe $P$ examples of the form $(x(i), y(i))$ where $x(i)$ is sampled independently and uniformly at random from $k$-sparse bitstrings in $\{0, 1\}^N$, and $y$ is given by $\epsilon_i + \langle x(i), x \rangle \pmod 2$, where $\epsilon_i$ is 1 with probability $\eta$ and 0 otherwise.

Given the examples $\{(x(i), y(i))\}_{i=1}^P$, the goal is to recover $x$.

The *SLPN assumption* is that for any $P = N^{(1-\rho)k/2}$ for constant $0 < \rho < 1$, and any sufficiently large inverse polynomial noise rate $\eta$, no $\mathrm{poly}(N)$-time algorithm can recover $x$ with high probability.

*Proof of Proposition B.2.* With probability at least $1 - (1 - 1/N)^N \geq \Omega(1)$, all of the variable tokens $x^{\pi(i)}$ for $i \leq N$ are masked. Independently, the number of unmasked tokens among the observation tokens $\mathcal{O}_j$ is distributed as $\mathrm{Bin}(P, 1 - \alpha)$, so by a Chernoff bound, with probability at least $1 - e^{-\Omega(P/N^2)} = 1 - 1/\mathrm{poly}(N)$ we have that at least $P/4N = \Omega(N \log N)$ observation tokens are unmasked. The masking problem in this case amounts to an instance of SLPN with input dimension $N$ and sample size in $[\Omega(N \log N), O(N^{0.49k})]$. Because of the lower bound on the sample size, prediction of $\mathbf{x}^M$ is information-theoretically possible. Because of the upper bound on the sample size, the SLPN assumption makes it computationally hard. As a result, estimating the posterior mean on any entry of $\mathbf{x}^M$ given the unmasked tokens is computationally hard as claimed. $\qquad \square$

## B.2. Additional example: random slab observations

**Example B.4** (Random slab observations). *Let $m = 2$ and $P = \gamma N^2$ for constant $\gamma > 0$. Fix slab width $\beta$ and vectors $z(1), \ldots, z(P)$ sampled independently from $\mathcal{N}(0, I)$. For each $j \in [P]$, define the corresponding observation $\mathcal{O}_j(x)$ to be deterministically 1 if $|\langle z(j), 2x - \mathbf{1} \rangle| \leq \beta \sqrt{N}$, and deterministically 0 otherwise.*

In (Alaoui & Gamarnik, 2024), it was shown that *stable* algorithms (Definition B.7), which encompass many powerful methods for statistical inference like low-degree polynomial estimators, MCMC, and algorithmic stochastic localization (Gamarnik, 2021), are unable to sample from the posterior distribution over a random bitstring conditioned on it satisfying $|\langle z(j), x \rangle| \leq \beta \sqrt{N}$ for any $\Theta(N)$ number of constraints $z(1), \ldots, z(P')$, provided $P'$ is not too large that the support of the posterior is empty. This ensemble is the well-studied *symmetric perceptron* (Aubin et al., 2019). The following is a direct reinterpretation of the result of (Alaoui & Gamarnik, 2024):

**Proposition B.5.** *Let $p_{\text{data}}$ be a L&O distribution with random slab observations as defined in Example B.4, with parameter $\gamma > 0$ and slab width $\beta > 0$. There exists a constant $c_\beta > 0$ such that for any absolute constant $0 < c < c_\beta$, if $1 - c_\beta N / 2P \leq \alpha \leq 1 - cN/P$ and $\gamma > c_\beta$, the following holds. Let $p'_{\text{data}}$ denote the distribution given by independently masking every coordinate in $p_{\text{data}}$ with probability $\alpha$. Then any $(1 - \tilde{\Omega}(1/\sqrt{N}))$-stable algorithm, even one not based on masked diffusion, which takes as input a sample $x'$ from $p'_{\text{data}}$ and, with probability $1 - o(1)$ outputs a Wasserstein-approximate[3] sample from $p_{\text{data}}$ conditioned on the unmasked tokens in $x'$, must run in super-polynomial time.*

The upshot of this is that any stable, polynomial-time masked diffusion sampler will, with non-negligible probability, encounter a computationally hard masking problem at some point during the reverse process.

For the proof, we first formally define the (planted) symmetric Ising perceptron model:

**Definition B.6.** *Let $\alpha, \beta > 0$. The planted symmetric Ising perceptron model is defined as follows:*

- Nature samples $\sigma$ uniformly at random from $\{\pm 1\}^N$

- For each $j = 1, \ldots, P = \lfloor \alpha N \rfloor$, we sample $z(j)$ independently from $\mathcal{N}(0, I_N)$ conditioned on satisfying $|\langle z(j), \sigma \rangle| \leq \beta \sqrt{N}$.

The goal is to sample from the posterior on $\sigma$ conditioned on these observations $\{z(i)\}_{i=1}^P$.

Next, we formalize the notion of *stable algorithms*.

**Definition B.7.** Given a matrix $Z \sim \mathcal{N}(0, 1)^{\otimes P \times N}$, define $Z_t = tZ + \sqrt{1 - t^2} Z'$ for independent $Z' \sim \mathcal{N}(0, 1)^{\otimes P \times N}$. A randomized algorithm $\mathcal{A}$ which takes as input $Z \in \mathbb{R}^{P \times N}$ and outputs an element of $\{\pm 1\}^N$ is said to be $t_N$-*stable* if $\lim_{N \to \infty} W_2(\text{law}(\mathcal{A}(Z)), \text{law}(\mathcal{A}(Z_t))) = 0$.

As discussed at depth in (Gamarnik, 2021), many algorithms like low-degree polynomial estimators and Langevin dynamics are stable.

**Theorem B.8** (Theorem 2.1 in (Alaoui & Gamarnik, 2024)[4]). *For any constant $\beta > 0$, there exists $c_\beta > 0$ such that the following holds for all constants $0 < \alpha < c_\beta$. For $t_N \leq 1 - \Omega(\log^2(n)/n^2)$, any $t_N$-stable randomized algorithm $\mathcal{A}$ which takes as input $Z = (z(1), \ldots, z(P))$ and outputs an element of $\{\pm 1\}^N$ will fail to sample from the posterior on $\sigma$ conditioned on $Z$ in the symmetric Ising perceptron model to Wasserstein error $o(\sqrt{N})$.*

*Proof of Proposition B.5.* By a union bound, with probability at least $1 - (1 - \alpha)N \geq 1 - c_\beta N^2/P \geq 1 - c_\beta/\gamma$ over a draw $x' \sim p'_{\text{data}}$, all of the $x^{\pi(i)}$ tokens are masked. The number of unmasked tokens in $x'$ among the observations $\mathcal{O}_j$ is distributed as $\text{Bin}(P, 1 - \alpha)$. By a Chernoff bound, this is in $[3cN/4, 3c_\beta N/4]$ with at least constant probability. The claim then follows immediately from Theorem B.8 above. $\square$

## B.3. Proof outline of Proposition 3.3

To understand the proof idea, we consider the case where all the latent tokens are masked and some of the observation tokens are unmasked. In this case, the prediction task reduces to learning to recover the latent tokens that are consistent

---

[3]Here the notion of approximation is $o(1)$-closeness in Wasserstein-2 distance.

[4]Note that while the theorem statement in (Alaoui & Gamarnik, 2024) refers to the non-planted version of the symmetric binary perceptron, the first step in their proof is to argue that these two models are mutually contiguous in the regime of interest.

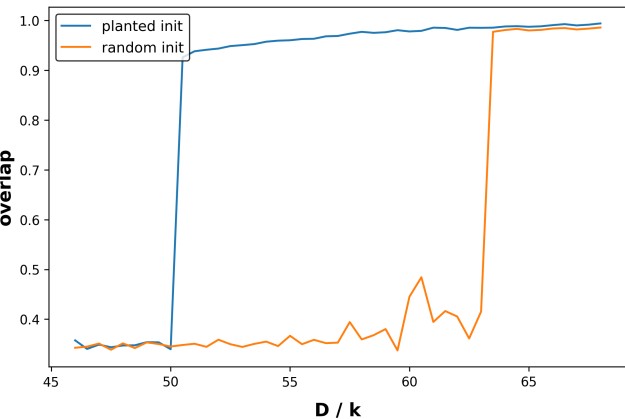

*Figure 4.* Overlap achieved by belief propagation initialized at ground truth versus random for planted CSP with $k = 3$, $m = 3$, and $g = \text{NAE}$, for $N = 10000$ and varying choices of average degree $D$. $D_{\text{KS}}/K$ can be shown analytically to be 64, consistent with the phase transition depicted. Plot suggests $D_{\text{cond}}/K \approx 50$. By Prop. 3.3 this implies a range of masking fractions at which $\Omega(1)$ fraction of masking problems are computationally hard.

with the observations. Intuitively, each observation provides some constraints and the task is to recover an assignment that satisfies the constraints. This is reminiscent of *Constraint Satisfaction Problems* (CSPs). Indeed, to show the hardness result, we use the rich theory developed for *planted* CSPs at the intersection of statistical physics and average-case complexity.

In a planted CSP, there is an unknown randomly sampled vector $y$ of length $N$ and, one is given randomly chosen Boolean constraints which $y$ is promised to satisfy, and the goal is to recover $y$ as best as possible (see Definition B.9). Prior works have shown the hardness of efficiently learning to solve the planted CSP problem (Krzakala & Zdeborová, 2009; Alaoui & Gamarnik, 2024). We show the hardness of masking problems in L&O distributions based on these results. Consider the ground truth latent tokens as the random vector $y$ and each observation as a constraint. In this case, the problem of learning to recover the latent tokens from the observation tokens reduces to recovery for the planted CSP.

There are precise predictions for the values of vocabulary size $m$ and the number of observations for which the information-theoretically best possible overlap and the best overlap achievable by any computationally efficient algorithm are different. We show that these predictions directly translate to predictions about when masking problems become computationally intractable:

As a simple example, let us consider sparse predicate observations with $k = 2$ and $g(x', x'') = \mathbf{1}[x' \neq x'']$. These can be formally related to the well-studied problem of *planted $m$-coloring*. In the planted $m$-coloring, a random graph of average degree $D$ is sampled consistent with an unknown vertex coloring and the goal is to estimate the coloring as well as possible (Krzakala & Zdeborová, 2009), as measured by the *overlap* of the output of the algorithm to the ground-truth coloring (see Definition B.9). As a corollary of our main result, we show that when all the latent tokens $x^{\pi(1)}, \ldots, x^{\pi(N)}$ are masked and a few unmasked observation tokens provide the information of the form $g(x^{\pi(i)}, x^{\pi(j)}) = \mathbf{1}[x^{\pi(i)} \neq x^{\pi(j)}]$ for $i, j \leq N$, then solving the masking problem can be reduced to solving planted coloring.

For planted $m$-coloring, when $m = 5$ the thresholds in Proposition 3.3 are given by $D_{\text{KS}}/2 = 16$ and $D_{\text{cond}}/2 \approx 13.23$ (Krzakala & Zdeborová, 2009) (the factor of 2 here is simply because the observations correspond to *ordered* subsets of size 2). For general predicates and arities, there is an established recipe for numerically computing $D_{\text{KS}}$ and $D_{\text{cond}}$ based on the behavior of the *belief propagation* algorithm (see the discussion in Appendix B.4). As an example, in Fig. 4, we execute this recipe for $m = 3$, $k = 3$, and $g$ given by the Not-All-Equal predicate $\text{NAE}(x', x'', x''') = 1 - \mathbf{1}[x' = x'' = x''']$ to obtain thresholds that can be plugged into Proposition 3.3.

**Additional examples of the hardness.** The above setup can also be generalized to capture *Bayesian constraint satisfaction problems* (Montanari, 2008; Liu et al., 2022), one notable example of which is the stochastic block model (Decelle et al., 2011). There are analogous predictions for the onset of hardness of inference, which can likewise be translated to hardness of masking problems for seemingly benign L&O distributions. In Appendix B.1 and B.2, we give two more examples of L&O distributions for which order-aware training is tractable yet order-agnostic training of the MDM is computationally

hard.

First, we consider L&O distributions whose observations are sparse, noisy parities in the latents and deduce hardness for order-agnostic training from the Sparse Learning Parity with Noise assumption (Alekhnovich, 2003). We then consider L&O distributions whose observations are *generalized linear models* in the latents, and deduce hardness for a large class of efficient algorithms from existing results on Lipschitz hardness (Alaoui & Gamarnik, 2024) for the symmetric binary perceptron (Aubin et al., 2019).

### B.4. Proof of Proposition 3.3: sparse predicate observations

Here we formally define the relevant notions needed to formalize our claim about hardness in Proposition 3.3.

**Definition B.9** (Planted CSPs). Given arity $k \in \mathbb{N}$, vocabulary/alphabet size $m \in \mathbb{N}$, predicate $g : \{1, \dots, m\}^k \to \{0, 1\}$, latent dimension $N$, and clause density $P/N$, the corresponding *planted constraint satisfaction problem* is defined as follows: Nature samples an unknown assignment $\sigma$ uniformly at random from $\{1, \dots, m\}^N$, and then for each ordered $k$-tuple $S$ of distinct elements from $[N]$, we observe the *clause* $S$ independently with probability $\phi/N^{k-1}$ if $g(\sigma|_S) = 1$.

To measure the quality of an algorithm for recovering $\sigma$ given the observations, define the *overlap* between an estimate $\hat{\sigma}$ and the ground truth $\sigma$ by $d(\sigma, \hat{\sigma}) \triangleq \min_{\pi \in \mathbb{S}_N} \sum_i \mathbf{1}[\sigma_i = \pi(\hat{\sigma}_i)]$ where $\mathbb{S}_N$ denotes the set of all permutations of $\{0, 1, \dots, N - 1\}$. Define the *average degree* to be $kP/N$, i.e. the expected number of variables that share at least one clause with a given variable.

We begin by defining the central algorithm driving statistical physics predictions about hardness for random constraint satisfaction problems: belief propagation (BP).

**Definition B.10** (BP update rules). Belief propagation is an algorithm that iteratively updates a set of *messages* $\{\mathrm{MS}_c^{i \to S}[t], \mathrm{MS}_c^{S \to i}[t]\}$, where $i, S$ range over all pairs of variable indices $i \in [N]$ and observations $S \ni i$. At time $t + 1$, the messages are computed via

$$\mathrm{MS}_c^{i \to S}[t + 1] \propto \prod_{T : i \in T \neq S} \mathrm{MS}_c^{T \to i}[t] \tag{4}$$

$$\mathrm{MS}_c^{S \to i}[t + 1] \propto \sum_{\overline{\sigma} \in \{1, \dots, m\}^{S \setminus i}} g(\overline{\sigma} \cup_i c) \prod_{j : i \neq j \in S} \mathrm{MS}_{\overline{\sigma}_j}^{j \to S}[t], \tag{5}$$

where $\overline{\sigma} \cup_i c \in \{1, \dots, m\}^S$ assigns $c$ to entry $i$ and $\overline{\sigma}$ to the remaining entries.

A set of messages can be used to estimate the marginals of the posterior on $\sigma$ conditioned on the observations as follows. The marginal on the $i$-th variable has probability mass function over $\{1, \dots, m\}$ proportional to $\{\prod_{T : i \in T} \mathrm{MS}_c^{T \to i}\}$. Given a set of marginals, a natural way to extract an estimate for $\sigma$ is to round to the color in $\{1, \dots, m\}$ at which the probability mass function is largest.

Throughout we will make the following assumption that ensures that the trivial messages $\mathrm{MS}_c^{i \to S} = 1/m$ and $\mathrm{MS}_c^{S \to i} = 1/m$ are a fixed point, sometimes called the *paramagnetic fixed point*, for the iteration above:

**Assumption B.11.** The quantity $\sum_{\overline{\sigma} \in \{1, \dots, m\}^{[k] \setminus i}} g(\overline{\sigma} \cup_i c)$ is constant across all $c \in \{1, \dots, m\}$ and $i \in [k]$.

**Definition B.12.** Given $k, m, g$, the *Kesten-Stigum* threshold $D_{\mathrm{KS}}$ is defined to be the largest average degree for which BP is locally stable around the paramagnetic fixed point, that is, starting from a small perturbation of the paramagnetic fixed point, it converges to the paramagnetic fixed point. More formally, $D_{\mathrm{KS}}$ is the largest average degree at which the Jacobian of the BP operator $\{\mathrm{MS}^{i \to S}[t]\} \mapsto \{\mathrm{MS}^{i \to S}[t + 1]\}$ has spectral radius less than 1.

The *condensation* threshold $D_{\mathrm{cond}}$ is defined to be the largest average degree at which the planted CSP ensemble and the following simple *null model* become mutually contiguous and thus statistically indistinguishable as $N \to \infty$. The null model is defined as follows: there is no single unknown assignment, but instead for every ordered subset $S$ of $k$ variables, Nature independently samples an unknown local assignment $\sigma_S \in \{1, \dots, m\}^S$, and the observation is included with probability $\phi/N^{k-1}$ if $g(\sigma_S) = 1$.

For $D_{\mathrm{cond}} < kP/N < D_{\mathrm{KS}}$, there exists some *other* fixed point of the BP operator whose marginals, once rounded to an assignment, achieves strictly higher overlap than does BP with messages initialized randomly. The prediction is that in this regime, no efficient algorithm can achieve optimal recovery (Krzakala & Zdeborová, 2009).

**Conjecture B.13** (1RSB cavity prediction). *Suppose $k, m, g$ satisfy Assumption B.11, and let $D_{\mathrm{KS}}$ and $D_{\mathrm{cond}}$ denote the associated Kesten-Stigum and condensation thresholds for the average degree. Then for all $P$ for which $D_{\mathrm{cond}} < kP/N < D_{\mathrm{KS}}$, the best overlap achieved by a computationally efficient algorithm for recovering $\sigma$ is strictly less than the best overlap achievable.*

*Proof of Proposition 3.3.* At masking fraction $\alpha$ satisfying the bounds in the Proposition, with probability at least $\alpha^N \geq (1 - \gamma^{-1} D_{\mathrm{KS}}/N^{k-1})^N \geq \Omega(1)$ we have that all tokens corresponding to latents $x_{\pi(i)}$ get masked. Independently of this, the number of unmasked tokens among the observation tokens $\mathcal{O}_S$ is distributed as $\mathrm{Bin}(N(N-1)\cdots(N-k+1), 1-\alpha)$, so by standard binomial tail bounds, with constant probability (depending on the gap between $D_{\mathrm{cond}}$ and $D_{\mathrm{KS}}$) this lies between $\gamma^{-1} D_{\mathrm{cond}} N/k$ and $\gamma^{-1} D_{\mathrm{KS}} N/k$. Furthermore, of these unmasked tokens in expectation $\gamma$ fraction of them correspond to observations for which the associated predicate evaluates to 1. Conditioned on the above events, the masking problem thus reduces exactly to inference for a planted constraint satisfaction problem at average degree $D_{\mathrm{cond}} < D < D_{\mathrm{KS}}$, from which the Proposition follows. $\square$

# C. Experimental details in Section 3

## C.1. Experimental details in Section 3.2

**$\pi$-learner configurations.** We consider two distributions of $\pi$ that interpolate between $\mathrm{Unif}(\mathbb{S}_L)$ where $\mathbb{S}_L$ denote the uniform distribution over all permutations of indices $\{0, 1, \ldots, L-1\}$ and the point mass at the identical distribution: (Closer) and (Much-closer). To construct those distributions, we start from the identity permutation and perform a certain number of random swapping operations. Since $L \log(L)$ number of swaps results in a distribution that is very close to $\mathrm{Unif}(\mathbb{S}_L)$ (Bormashenko, 2011), we use $L/10$ and $\sqrt{L}$ swaps to construct the (Closer) and (Much-closer) distributions, respectively. For consistency, we repeat this sampling process three times.

**Model and training configurations.** As explained in Section 3.2, to evaluate the scaling law of the $\pi$-learner, we can simply adapt the autoregressive training setup (a transformer with causal attention) by modifying the input to $\pi(x_0)$ and using a learnable positional embedding layer instead of RoPE. We borrow the training configurations from (Nie et al., 2024), which are also consistent with the TinyLlama (Zhang et al., 2024) configurations. In particular, we use AdamW optimizer (Loshchilov & Hutter, 2017), setting $\beta_1 = 0.9$, $\beta_2 = 0.95$, and a weight decay of $0.1$ and $L = 2048$. A cosine learning rate schedule is applied, with a maximum learning rate of $4 \times 10^{-4}$ and a minimum learning rate of $4 \times 10^{-5}$. We also note that **unless otherwise specified, we maintain the same training configuration throughout the paper.**

**Examining scaling laws.** We conduct IsoFLOP analysis (Hoffmann et al., 2022). For a given number of FLOPs $C$, by varying the number of non-embedding parameters of transformers, we set the iteration numbers so that the total number of tokens observed by the model during training equals $C/6N$, following prior studies (Hoffmann et al., 2022; Kaplan et al., 2020). We then select the smallest validation loss and set it as a data point.

## C.2. Experimental details in Section 3.3

### C.2.1. EXPERIMENT ON L&O-NAE-SAT DISTRIBUTION

We consider the L&O-NAE-SAT distribution with $(N, P) = (20, 280)$. For each example sequence from L&O-NAE-SAT, we pad the last 212 tokens with an additional token value of 2. We employ a 19M MDM with RoPE and a maximum sequence length of 512. Then, this MDM is trained for $2 \times 10^3$ iterations. To attain a proxy MDM for the Bayes optimal predictor, we further train it for $5 \times 10^4$ iterations.

To measure the error across different tasks, we consider the following setup. For each $\ell \in [1, N-1]$, we randomly mask $\ell$ tokens in the latent positions and $\ell \times (P/N)$ tokens in the observed positions. Across all masked prediction positions, $\ell(1 + P/N)$, we measure the error for each position. For certainty, we repeat this process 1000 times. The result in Figure 2 corresponds to the case when $\ell = 11$, and we observe the same tendency for other values of $\ell$.

### C.2.2. EXPERIMENT ON TEXT DATA

We take a 170M MDM pretrained with text data for a baseline model. To measure the performance imbalance between likelihood modeling tasks

$$\mathbb{E}_{x_0 \sim p_{\text{data}}} \left[ \sum_{i=0}^{L-1} \log p_\theta \left( x_0^{\pi(i)} \Big| x_0[\pi\{i, \ldots, L-1\}] \right) \right].$$

As done in the experiments in Section 3.2, we sample $\pi$s from three different distributions: $\text{Unif}(\mathbb{S}_L)$, (Closer), the point mass of identical distribution. For each case, we calculate the expectation over 1024 samples of $x_0 \sim p_{\text{data}}$.

# D. Experimental details in Section 4

## D.1. Experimental details in Section 4.2

### D.1.1. EXPERIMENT ON L&O-NAE-SAT DISTRIBUTION

We consider five instances of L&O-NAE-SAT: $(N, P) = (25, 275), (30, 270), (40, 260), (50, 250), (100, 200)$. For each distribution, we train a 19M MDM and measure the accuracy difference between vanilla inference and adaptive inference using Top-K probability margin.

### D.1.2. EXPERIMENT ON TEXT DATA

**Top-$K$ probability margin sampler with temperature.**   To modify our inference for text data modeling, which does not have a determined answer, we found that adding a certain level of temperature to the oracle is useful. This is because Top-$K$ probability margin or Top-$K$ often leads to greedy sampling, which harms the diversity (entropy) of the generated samples. Therefore, we consider a variant of the oracle as follows, incorporating a noise term $\epsilon$:

$$\mathcal{F}(\theta, x_t) = \text{Top } K \left( |p_\theta(x^i = j_1|x_t) - p_\theta(x^i = j_2|x_t)| + \epsilon \right).$$

Note that this approach has also been employed for unconditional sampling (Wang et al., 2024; Zheng et al., 2023).

**Generative perplexity and entropy.**   We employ a 1.1B MDM pretrained on text data as a baseline. For each sampling step, we unconditionally generate samples using both vanilla and adaptive inference. Next, we calculate the likelihood using LLama2-7B as a baseline large language model. Moreover, we denote the entropy of a generated sample $x$ as $\sum p_i \log p_i$, where $p_i = \#\{x^i = i\}/L$.

## D.2. Experimental details on Sudoku and Zebra puzzles

**Dataset.**   For both Sudoku and Zebra puzzles, we use the dataset provided in Shah et al. (2024) to train our model. To evaluate our model on the same difficulty tasks, we use the test dataset proposed in Shah et al. (2024). This dataset is created by filtering the puzzles from (Radcliffe, 2020) that can be solved using a fixed list of 7 strategies. To create a hard dataset to evaluate easy-to-hard generalization, we use the remaining puzzles from (Radcliffe, 2020) as they either require a new strategy unseen during the training and/or require backtracking. The hard dataset contains around 1M Sudoku puzzles.

**Model, training, and inference.**   For the Sudoku dataset, we use 6M GPT-2 model and for the Zebra dataset, we use 19M model but instead of causal attention, we use complete bidirectional attention. We set the learning rate to 0.001 with batch size 128 to train the model for 300 epochs. For the inference, we use 50 reverse sampling steps using the appropriate strategy. Additionally, we add Gumbel noise with a coefficient of 0.5 to the MDM inference oracle $\mathcal{F}$.

# E. Omitted proofs

*Proof of Proposition 2.1.* We first re-state the Proposition 3.1 from (Zheng et al., 2024). To clarify, (Zheng et al., 2024) generally considers the case beyond the time-embedding denoising network $p_\theta$.

**Proposition E.1** (Proposition 3.1 of (Zheng et al., 2024))**.** *For clean data $x_0$, let $\tilde{q}(x(n) \mid x_0)$ be the discrete forward process that randomly and uniformly masks $n$ tokens of $x_0$. Suppose $\alpha_0 = 0$ and $\alpha_1 = 1$. Then the MDM training loss (1) can be reformulated as*

$$\mathcal{L}_\theta = -\sum_{n=1}^{L} \mathbb{E}_{x(n) \sim \tilde{q}(\cdot|x_0)} \left[ \frac{1}{n} \sum_{\ell: x(n)^\ell = m} \mathbf{e}_{x_0^\ell} \log p_\theta(x^\ell \mid x(n)) \right]. \tag{6}$$

To obtain an alternative formulation of (6), we expand the expectation $x(n) \sim \tilde{q}(\cdot \mid x_0)$. Since there are total $L$ positions of $x_0$, we have the probability assigned for each $x(n)$ equals $1/\binom{L}{n}$. Therefore,

$$
\begin{aligned}
\mathcal{L}_\theta &= -\sum_{n=1}^{L} \mathbb{E}_{x(n)\sim\tilde{q}(\cdot|x_0)} \left[ \frac{1}{n} \sum_{\ell:x(n)^\ell = m} \mathbf{e}_{x_0^\ell} \log p_\theta(x^\ell \mid x(n)) \right] \\
&= - \sum_{M\in[L], i\in M} \frac{1}{\binom{L}{|M|}} \times \frac{1}{|M|} \mathbf{e}_{x_0^\ell} \log p_\theta(x^\ell \mid x[M]) \\
&= - \sum_{M\in[L], i\in M} \frac{1}{\binom{L}{|M|}} \times \frac{1}{|M|} \log p_\theta(x_0^\ell \mid x[M]) \\
&= - \sum_{M\in[L], i\in M} \frac{1}{L\binom{L-1}{|M|-1}} \log p_\theta(x_0^\ell \mid x[M]).
\end{aligned}
$$

$\square$

*Reformulating the MDM loss with $\pi$-learner s.* In this paragraph, we provide the proof of

$$
-\frac{1}{L} \sum_{M\subseteq[L], i\in M} \frac{1}{\binom{L-1}{|M|-1}} \mathop{\mathbb{E}}_{x_0\sim p_{data}} [\log p_\theta(x_0^i|x_0[M])] = -\mathbb{E}_{\pi\sim\mathrm{Unif}(\mathbb{S}_L), x_0\sim p_{data}} \left[ \sum_{i=0}^{L-1} \log p_\theta\left( x_0^{\pi(i)} \Big| x_0[\pi\{i, \ldots, L-1\}] \right) \right].
$$

Alternatively, we will demonstrate that

$$
-\frac{1}{L} \sum_{M\subseteq[L], i\in M} \frac{1}{\binom{L-1}{|M|-1}} \log p_\theta(x_0^i|x_0[M]) = -\mathbb{E}_{\pi\sim\mathrm{Unif}(\mathbb{S}_L)} \left[ \sum_{i=0}^{L-1} \log p_\theta\left( x_0^{\pi(i)} \Big| x_0[\pi\{i, \ldots, L-1\}] \right) \right]
$$

holds for every $x_0$. Note that

$$
\begin{aligned}
&\mathbb{E}_{\pi\sim\mathrm{Unif}(\mathbb{S}_L)} \left[ \sum_{i=0}^{L-1} \log p_\theta\left( x_0^{\pi(i)} \Big| x_0[\pi\{i, \ldots, L-1\}] \right) \right] \\
&= \frac{1}{L!} \sum_{\pi\in\mathbb{S}_L} \sum_{j=0}^{L-1} \log p_\theta\left( x_0^{\pi(j)} \Big| x_0[\pi\{j, \ldots, L-1\}] \right).
\end{aligned}
$$

Next, by regarding $\pi\{j, \ldots, L-1\} = \{\pi(j), \ldots, \pi(L-1)\} = M \subseteq [L]$ and $\pi(j) = i$ in the equation (1), we count the number of $\pi \in S_L$ that induces a specific term $\log p_\theta(x_0^i|x_0[M])$. For a given $M \in [L]$ and $i \in M$, $\pi$ must satisfy

$$
\pi(j) = i, \quad \{\pi(j), \ldots, \pi(L-1)\} = M.
$$

The number of $\pi$ that satisfies above is $(L-|M|)! \times (|M|-1)!$. Finally, the following calculation concludes the proof.

$$
\begin{aligned}
&\mathbb{E}_{\pi\sim\mathrm{Unif}(\mathbb{S}_L)} \left[ \sum_{i=0}^{L-1} \log p_\theta\left( x_0^{\pi(i)} \Big| x_0[\pi\{i, \ldots, L-1\}] \right) \right] \\
&= \frac{1}{L!} \sum_{\pi\in S_L} \sum_{j=0}^{L-1} \log p_\theta\left( x_0^{\pi(j)} \Big| x_0[\pi\{j, \ldots, L-1\}] \right) \\
&= \frac{1}{L!} \sum_{|M|\in[L], i\in M} \left[ \log p_\theta(x_0^i|x_0[M]) \times (L-1-|M|)! \times (|M|-1)! \right] \\
&= \frac{1}{L} \sum_{|M|\in[L], i\in M} \frac{1}{\binom{L-1}{|M|-1}} \times \log p_\theta(x_0^i|x_0[M]).
\end{aligned}
$$

$\square$

