# OpenReview forum: "Train for the Worst, Plan for the Best: Understanding Token Ordering in Masked Diffusions"
_ICML.cc/2025/Conference — ICML 2025 oral_

### Official Review · Reviewer_oBo5 · 2025-03-03

**Overall Recommendation:** 3

**Summary:**

The authors first show masked diffusion models (MDMs) indeed train on computationally intractable subproblems compared to their autoregressive counterparts. Then an adaptive Top-K probability margin inference strategy is proposed to sidestep hard subproblems that are not properly learned in the training time. The proposed inference strategy has proven to be effective in the Sudoku puzzle task.

**Claims And Evidence:**

The claim of "Complexity at training time" of MDMs is well shown both theoretically and empirically on text data.
The claim of "Planning for the best" of MDMs is less persuasive. For instance, the Top-K probability margin inference strategy proves effective for Sudoku puzzles but does not work for the Zebra puzzle, and its performance on text data remains unknown.

**Essential References Not Discussed:**

No.

**Ethics Expertise Needed:**

["Other expertise"]

**Experimental Designs Or Analyses:**

The experimental designs in Sections 3 and 4 are mostly correct.

**Methods And Evaluation Criteria:**

The evaluation of the imbalanced subproblems during the training time of MDMs and the proposed adaptive inference strategy makes sense in general.

**Other Comments Or Suggestions:**

N.A.

**Other Strengths And Weaknesses:**

Strengths:
- The analysis of why "MDMs train on hard problems" is sufficient and helps those unfamiliar with MDMs understand their characteristics.
- The Top-K probability margin inference strategy is intuitive and effective on Sudoku.

Weaknesses:
- The proposed Top-K probability margin inference strategy proves effective only for Sudoku puzzles but not for text data, which raises concerns about the technical contribution to the general text domain and diffusion LLMs.

**Questions For Authors:**

Regarding adaptive Top-K probability margins: As it is stated "When multiple values have similar probabilities at a position, Top-K probability margin will provide a better estimate of the uncertainty of a position". Does this mean the Top-K probability margin strategy is only useful when there are multiple possible values for a given position? If so, why is the Top-K probability margin so effective for Sudoku as each position has only one solution in a standard Sudoku puzzle. I would appreciate further clarification on this.

**Relation To Broader Scientific Literature:**

The findings in the paper may be useful in building the next-generation LLMs based on the diffusion model.

**Theoretical Claims:**

I reviewed the proofs in Section 2 and they seemed correct to me.

---

> ### Author Rebuttal · Authors · 2025-03-31
>
> We appreciate the reviewer’s valuable questions and comments. Below, we address the main concerns.
>
> ## (1) Further experiments on text data
> In response to the reviewer’s comments, we ran additional experiments and found that **Top-k margin indeed outperforms Top-k on challenging code and math tasks**.
>
> Specifically, to examine the effect of different inference strategies on text evaluation tasks, we adapted LLaDA, the 8B MDM model from [1]. We compare three strategies: **Vanilla, Topk, Topk-margin**.  The results are presented below.
>
> | **Sampler**      | HumanEval-Single | HumanEval-Multi | HumanEval-Split | Math        | MMLU-Pro | ROCStories |
> |------------------|-------------------------|------------------------|------------------------|--------|----------|-------------|
> | **Vanilla**      | 31.8%                   | 16.5%                  | 14.2%                  | 28.5%  | 33.2%    | 21.23%      |
> | **Top-k**        | 32.9%                   | 20.8%                  | 18.4%                  | 31.3%  | **36.5%** | 21.10%      |
> | **Top-k Margin** | **33.5%**               | **25.4%**              | **22.3%**              | **34.3%** | 35.4%    | **21.41%**   |
>
>
> As shown in the table, both Top-k and Top-k Margin consistently outperform the Vanilla MDM inference, underscoring the importance of adaptively selecting the decoding order to avoid harder problem instances. Notably, in more challenging tasks, such as HumanEval-Multiline, HumanEval-Split Line, and Math, Top-k Margin shows a clear advantage over Top-k.
>
> This is because, particularly in coding and math problems where the fixed answer exists, selecting the correct intermediate token during inference is critical. Moreover, the Top-k margin offers a more reliable estimate of uncertainty when multiple tokens have similar probabilities—a common scenario in these challenging tasks. These results reinforce our claim in Section 4.1: Top-k Margin serves as a better proxy for positional uncertainty than Top-k in such cases. **These results also further highlight the potential of the Top-k Margin strategy for challenging infilling tasks.**
>
> **We also emphasize that our main contribution lies in the fundamental understanding of token ordering in MDM, rather than in proposing a superior inference strategy.** Nevertheless, as demonstrated in the experimental results on Sudoku puzzles and math/coding tasks, our proposed Top-$k$ Margin inference shows promising potential compared to Top-$k$ in certain scenarios, highlighting practical implications of our work. **Given that the main weakness raised was that we did not demonstrate effectiveness for our inference strategy on text data, we hope the reviewer will consider raising their score.**
>
> ## (2) Clarification about Top-K probability margin
> The Top-K probability margin strategy is not only useful when there are multiple possible correct values for a given position, but can also be effective when there is a single correct value. The reason is that the strategy is used for the **distribution $p_\theta$ estimated by the model** rather than for the **true distribution $p_{data}$.**
>
> To understand this distinction, consider the case of Sudoku. Recall that $x_t$ denotes the sequence at decoding time $t$ – in the case of Sudoku, this is the partially filled Sudoku puzzle. The posterior data distribution at the $i$-th location, given the partially filled puzzle, is denoted by $p_{data}(x^i | x_t)$. Since we are dealing with puzzles that have unique solutions, the reviewer is correct that $p_{data}(x^i = v | x_t) = 1$ at the correct value $v$ and 0 otherwise.
>
> However, the Top-K probability margin strategy doesn’t rely on $p_{data}(x^i | x_t)$ but instead uses $p_{\theta}(x^i | x_t)$ during the adaptive inference. As explained in Section 3, the learned posterior can be quite different from the true posterior. Intuitively, $p_{\theta}(x^i | x_t)$ reflects the model’s uncertainty about the correct value at the $i$-th location. When the model is unsure, it assigns high probabilities to a few candidate values which we refer to as the possible values at that position.
>
> The Top-K probability margin is effective in Sudoku because situations often arise where the model is uncertain between two or more possible values — say, $v_1$ and $v_2$ — assigning high probabilities to both. In these cases, the top probability $\max( p_{\theta}(x^i = v_1 | x_t), p_{\theta}(x^i = v_2 | x_t) )$ does not provide a reliable estimate of whether the model knows the correct value at the ith location. However, the Top-K probability margin $| p_{\theta}(x^i = v_1 | x_t) - p_{\theta}(x^i = v_2 | x_t) |$ serves as a more effective measure of the model’s uncertainty.

---

### Official Review · Reviewer_qGCY · 2025-03-07

**Overall Recommendation:** 4

**Summary:**

The paper takes a close look at training and inference of masked diffusion models (MDMs), which are a type of discrete diffusion models where the noising process consists of randomly “masking” tokens until all tokens are masked, and training a model to reverse this degradation process.
The paper claims that this order-agnostic training is inherently harder than left-to-right next-token prediction, that this difference can at least partly explain the gap in performance between AR models and MDMs, and that we can leverage the trained MDM to avoid difficult denoising orders. To this end, a novel sampling adapter for diffusion models is introduced, which significantly improves performance both on generative PPL for language modeling, as well as solving logical puzzles like Sudoku or Zebra puzzles.

**Claims And Evidence:**

The question of whether or not “the benefits of inference flexibility for MDMs is enough to outweigh the drawbacks of training complexity” is claimed to be answered “in the affirmative.” While this does seem to be the case for solving Sudoku puzzles, the answer is not as clear for language modeling. Unfortunately, there is no comparison between AR sampling from an AR model (and/or left-to-right sampling from a MDM as a non-adaptive baseline) and adaptive sampling with the proposed sampling algorithm from a MDM. If AR models still have better sample quality than MDMs with adaptive sampling, which is rather plausible, then the answer to the original question would unfortunately change to “it depends.”

L35 ff., Col. 2: I cannot verify the claim that “MDMs can actually be used to decode in any order” based on the provided reference. As far as I can tell, the cited paper does not conduct any experiments about the order-sensitivity of MDMs. Of course, in theory MDMs can decode in any order, but whether or not this is true in practice is a different question.

**Essential References Not Discussed:**

It is known that auto-regressive models trained with teacher forcing face some fundamental challenges [1], which may (at least partly) be to blame for their poor performance on Sudoku. There has also been work on the directionality of AR language models, finding that there seems to be a slight but consistent left-to-right bias in human language. Finally, a recent study (concurrent work) has applied image-based diffusion models to solving Sudoku [3].

All of these are not strictly necessary to cite, but may help tie the results into the broader literature. It is for the authors to decide whether or not to include them.

- [1] Bachmann & Nagarajan, 2024. https://arxiv.org/abs/2403.06963
- [2] Papadopoulos et al., 2024. https://arxiv.org/abs/2401.17505
- [3] Wewer et al., 2025. https://arxiv.org/abs/2502.21075

**Experimental Designs Or Analyses:**

Sampling adapters often decrease the diversity of generated samples, which in extreme cases can lead to a collapse of the distribution. For the proposed adaptive inference there does indeed seem to be somewhat of a decrease in entropy, which warrants providing some qualitative examples in the appendix to prove that no catastrophic collapse is occurring.

**Methods And Evaluation Criteria:**

Besides what is mentioned above, the methods and evaluation criteria seem sound.

**Other Comments Or Suggestions:**

Nits:
- L104: $e_{x_0^i}$ should be bold.
- Figure 1 (bottom): 2nd line, 2nd step; mask tokens should presumably have a black background.
- Definition 3.1: As stated, the vocabulary size is $q+1$, and $p_{data}$ should presumably be $\{1, \dots, q\}^L$. Also, lowercase $n$ is not defined and presumably refers to uppercase $N$.
- L216: I think it should be $g(x | S)$, not $g(x |_S)$.
- Figure 3 is not referenced in the text.
- Conjecture B.13 does not have a citation.
- Def. 3.1: Overloaded notation: $\pi$ is used for both permutation and latent distribution.

**Other Strengths And Weaknesses:**

The paper makes an important observation on how the infilling order in masked diffusion can have a major effect on both upstream and downstream performance. While the theoretical part seems a bit shaky, the empirical evidence is convincing.

Besides the theoretical part, the main weakness of the paper lies in lacking scientific rigor and a tendency to overstate or misrepresent the actual results. For example, the phrase “train for the worst” seems to imply that it is optimal to train on all possible permutations jointly, but this claim is not tested in the paper. Similarly, “planning for the best” implies that there is some sort of planning involved, which there isn’t (the term “planning”, in the context of Machine Learning, generally refers to the act of looking ahead of and beyond the immediate next step). In actuality, the paper proposes a sampling adapter, the likes of which are ubiquitous for autoregressive models. Applying this idea to discrete diffusion models is a novel and valuable contribution, and obfuscating it through a misleading title is not necessary.

Despite the concerns about soundness and in light of the strong empirical results, I am inclined to recommend an accepting decision and will be happy to update my score if these concerns can be addressed.

**Questions For Authors:**

1. L189 ff.: If The observations are cryptographic hash functions, is it not true that the observations themselves are also not efficiently learnable? Or, put differently, are cryptographic hash functions efficiently learnable? It seems to me like the answer would be no, in which case this example is not only worst-case, but also violates our assumption.
2. How is the “hardness” of a $\pi$-learner measured, and what is the exact hardness of the quoted “$\pi$-learner-much_closer”, “$\pi$-learner-closer”, and “$\pi$-learner-unif”? How does this compare to the average-case hardness, which would apply for MDM?
3. If different orderings have different inherent difficulty, and MDM is trained on all of them jointly, how does MDM perform on inherently easy (e.g. left-to-right) orders? According to the claims in the paper, we would expect this to be quite close to AR models and would be a nice addition to Figure 2.
4. As an alternative to the top-k probability margin: Could the per-token entropy be a better proxy for uncertainty?

**Relation To Broader Scientific Literature:**

The paper shines a light on the discrepancy between AR models and discrete (masked) diffusion models, which has been observed many times in the literature. It also proposes a novel adaptive sampling technique that drastically improves performance both on text and logic puzzles. Both of these are valuable contributions to the literature of masked diffusion models.

**Theoretical Claims:**

The paper makes the following two core theoretical claims.

Claim 1:
> we provide theoretical [...] evidence that the overhead imposed by training complexity quantifiably impacts MDMs’ performance.

Claim 2:
> We prove that even for simple, benign models of data, there are noise levels at which a large fraction, but not all, of the corresponding subproblems that MDMs must solve are computationally intractable.

While the proposed L&O distributions seem very contrived (and calling it a “benign” model of data is, IMO, a stretch when it is known that these problems can be computationally hard), Proposition 3.3 does seem correct. The fact that it does require assuming the “1RSP cavity prediction” conjecture to be true may need to be highlighted more prominently (although I’m personally not familiar with this conjecture and cannot judge whether it is generally accepted to make this assumption). However, the original statement (claim 2) seems somewhat trivially true: Of course there exist distributions where for some noise levels (namely, when all tokens are masked) solving the corresponding subproblem is computationally hard and for some noise levels it is easy (namely, when all tokens are unmasked). A more useful proposition may be that some _orders_ of filling in the missing tokens are harder than others. How might Proposition 3.3 imply that some infilling orders are computationally hard?

Similarly, and from what I can tell, the statement(s) proved in Appendix B.2 (Proposition B.5) does/do not necessarily imply that “order-aware training is tractable yet order-agnostic training is hard”. As far as I understand, what is shown is that for some noise level, finding the solution is computationally hard. If this indeed implies that some _orders_ are more difficult than others, further explanation and/or proof is warranted.

All in all, the original claim (claim 1) of “providing theoretical evidence that [...] training complexity quantifiably impacts MDMs’ performance” is either overstated or stands on shaky ground. The provided hardness proofs seem correct upon skimming, but from what I can tell they do not lead to a conclusion this strong. It is possible that I'm simply not seeing the final logical step, in which case it should be a simple fix of adding a more detailed explanation that leads to the final conclusion.

---

> ### Author Rebuttal · Authors · 2025-03-31
>
> We thank the reviewer for their insightful review and address the comments below.
>
> ## Soundness of theoretical claims
> There are several misunderstandings, so we'd like to clarify them. The statement "There exist distributions where for some noise levels (namely, when all tokens are masked) solving the corresponding subproblem is computationally hard and for some noise levels it is easy (namely, when all tokens are unmasked).”  **is incorrect**. The subproblem we investigate is to estimate the *coordinate-wise marginals* of the posterior distribution, not **full posterior sampling.** If all tokens are masked, full posterior can indeed be hard: take any hard-to-sample distribution. However, estimating marginals isn't necessarily difficult.
>
> For example, take a hard-to-sample Ising model with density $\propto e^{-x^TAx},x\in \\{-1,1\\}^n$. Thanks to sign-symmetry, marginals are unbiased, so estimating them is trivial—even if full sampling is hard! Our theory emphasizes scenarios where some intermediate masking fractions are computationally harder than either extreme (fully masked or unmasked).
>
> In vanilla MDM inference, at each step, a random subset of positions is selected to be unmasked. Consequently, the masking patterns encountered correspond to randomly sampled mask indices---precisely the setting considered in Proposition 3.3! In contrast, decoding in a fixed left-to-right order (as in ARM) leads to encountering only left-to-right subproblems. **These hopefully address the confusion on why our results imply the hardness of sampling under certain token orderings.**
>
> Regarding 1RSB, it is a widely accepted conjecture from statistical physics, with extensive literature support. For an introduction, see “Notes on computational-to-statistical gaps” by Bandeira, Perry, and Wein.
>
> ## Comparison to AR baseline
> We ran generative perplexity experiments using an ARM baseline. An 1.1B ARM achieved perplexity 11.745, lower than the 1.1B MDM’s 13.396 with adaptive inference. We acknowledge that our phrasing may have given the impression that adaptive MDM inference outperforms ARM. **However, our claim is more nuanced: adaptive MDM inference helps avoid hard problem instances.** Absolute ARM performance isn't directly relevant. To clarify, for Sudoku puzzles, we included ARM to demonstrate adaptive MDM’s advantage through flexible reasoning orders.
>
> ## Other comments
> - *On decoding in any order in MDM*: For “MDMs can actually decode in any order”, we only meant that theoretically, when all the infilling problems are perfectly solved, any-order decoding matches the true likelihood. We'll update the PDF to make it clear.
> - *On the title*: For “train for the worst”, we never claimed optimality of training over all permutations, only the benefits of training in fixed order. For “plan for the best,” the reviewer is conflating “sampling adapters, the likes of which are ubiquitous for auto-regressive models” with what we do. For AR, the adapter has nothing to do with decoding *order*, which is left-to-right by default. The reason we call it planning is that MDMs can decide (plan) which token position to decode at each step.
> - *Entropy dropping*: While there is indeed an entropy decrease on adaptive MDM inference, this drop is negligible. To contextualize this, we measured entropy using the SlimPajama dataset: *average (5.10) and 0.45 quantile (4.85)*. These demonstrate minimal entropy reduction during adaptive inference, thus insignificantly impacting text generation performance.
>
> ## Questions
> 1) **Learnability of hash functions**: This nuanced issue is covered in “Cryptography in NC0” by Applebaum et al., demonstrating cryptographic primitives implemented via constant-depth circuits can be polynomially learnable.
> 2) **$\pi$-learner**: Each $\pi$-learner learns sequences according to $\pi$ and is modeled via causal Transformers trained on permuted data. A higher likelihood indicates easier learning. Average-case hardness, applied to MDM, involves uniformly sampling permutations $\pi$, resulting in a lower likelihood compared to fixed left-to-right ordering (Fig 2, left, green line). Due to character limits, we kindly ask the reviewer to refer to the 'experimental setup' paragraph in Section 3.2 for more details. We are happy to clarify further on the discussion page.
> 3) **Performance on left-to-right sampling**: We observed catastrophic collapse during **left-to-right MDM sampling**, with entropy dropping (~0.28). This is because MDM wasn't explicitly trained for left-to-right order. This result underscores the importance of adaptive inference strategies, where the model selects unmasking positions based on logit-derived uncertainty rather than on a prefixed order.
> 4) **Per-token entropy strategy**: While a more natural measure would be per-token entropy, the only reason we went with top-k margin was its efficiency. In preliminary experiments, we also tried per-token entropy, but the performance difference was negligible.

---

> > ### Comment · Reviewer_qGCY · 2025-04-03
> >
> > Thank you to the authors for their detailed response.
> >
> > - **Soundness of theoretical claims**:
> > Thank you for clarifying this misunderstanding. I now believe that the theoretical results indeed show that masked diffusion models face computationally hard sub-problems on some models of data. Perhaps this elaboration could be included in a future version of the paper in order to guide the reader and avoid any confusion.
> > - **Comparison to AR baseline**:
> > Indeed, the results shown in the paper are more nuanced than that adaptive MDMs generally outperforms ARMs, and I believe the phrasing should be adapted and clarified accordingly. For example, L45 Col. 2 ("Training for the worst") comes across as if the theoretically quantifiable impact on training performance applies in general, including to language modeling, which is where MDM are featured most prominently. Instead, it should be clarified that this is provably true on some special (toy) models of data. Similarly, the next paragraph ("Planning for the best") comes across as if "the benefits of inference flexibility for MDMs enough to
> > outweigh the drawbacks" _in general_. Instead, this is only true for some models of data, including the toy models but also Sudoku puzzle solving. Indeed, and as the authors admit, it does not close the gap to ARM on language modeling, which is a caveat worth highlighting prominently (including gen. PPL numbers as provided in the rebuttal). It is important to realize that being upfront with caveats and limitations does not diminish the contributions of the paper, but actually improves the clarity and scientific rigor of the writing.
> > - **Sampling adapters**: The proposed adaptive inference is arguably still a sampling adapter. Instead of sampling $z\_{t-1} \sim p\_\theta(z\_{t-1} | z\_t)$, we sample from a modified distribution $\tilde{p}\_\theta(z\_{t-1} | z\_t)$, where $\tilde{p}\_\theta$ is a function of $p\_\theta$. Again, drawing this parallel does not diminish the contributions, but actually improves the paper by appropriately tying it into the existing literature.
> >
> > Given that my main concern regarding theoretical soundness has been addressed, I will increase my score from 3 (weak accept) to 4 (accept), while also urging the authors to improve phrasing and messaging as outlined above and in my initial review in order to avoid confusion and misconceptions. As I said in my original review, there is no need to conflate and obfuscate since the presented results are strong on their own.
> >
> > ---
> >
> > Nits:
> > - Providing a reference on the 1RSB conjecture will help make this paper more accessible to the general machine learning community. The same goes for polynomial learnability of cryptographic hash functions.
> > - Entropy decrease on language generation is expected and reasonably small, it should be included in the paper along with gen. PPL numbers.
> > - Catastrophic collapse on left-to-right sampling is interesting and important to highlight (esp. given the claim that "MDMs can decode in any order"). However, an entropy drop by 0.28 (as opposed to "to 0.28") does not indicate catastrophic collapse. Providing qualitative examples (in the appendix) can help give an idea of the nature and extent of the collapse.

---

> > > ### Author Response · Authors · 2025-04-09
> > >
> > > We appreciate that the reviewer found our rebuttal clarifying. For the further suggestions, we will make sure to include those in a new version.

---

### Official Review · Reviewer_XERQ · 2025-03-10

**Overall Recommendation:** 4

**Summary:**

The main contribution of the paper is the use of theoretical arguments and carefully designed experiments to show the following:

1. The complexity of training Masked Diffusion Models (MDMs) is higher than Auto-regressive Models (ARMs).

2. The flexibility of any-order decoding offered by MDMs helps it to perform better than ARMs on specific kinds of data distributions, especially the ones where some (data/instance dependent) positions in the sequence contain harder sub-problems than other positions.

The paper also introduces a new decoding strategy called the "top-k probability margin"-based strategy, which picks the next token to decode based on the margin between top-2 vocabulary items at any specific position.

**Claims And Evidence:**

## 1. The inference flexibility provided by the any-order decoding in MDMs overweights the drawbacks introduced by training complexity.

Nie et. al. (2024) already demonstrated through scaling law curves that the complexity of training MDMs is higher than ARMs. The paper provides some theoretical insight into the phenomenon. Their empirical experiments on text (Figure 2) re-confirm the observations made in Nie et. al. (2024).  Zheng et. al. (2024b) demonstrated that the logits produced by MDMs have useful information for selecting the positions to unmask during inference. In summary, Nie et. al. and Zheng et. al. together have already demonstrated the main claim of this paper. Therefore, in my view, the main contribution of this paper is the use of theoretical arguments and carefully designed synthetic experiments to drive the point home, which the paper does well.

## 2. The proposed top-k probability margin strategy for sampling performs better than the top-k strategy proposed in Zhen et. al. 2024b.

The proposed top-k prob. margin-based sampling strategy is only demonstrated to be better than top-k (from Zhen et. al. 2024b) on Sudoku puzzles. Both top-k and top-k prob. margin work similarly for Zebra puzzles (Table 3). Moreover, it is not clear if there is any advantage to using the top-k prob. margin-based strategy on real data like the text data used in the paper (Figure 4 does not compare the two adaptive sampling strategies). Therefore, I find the use of top-k prob. margin-based strategy to be not well justified and a possible area to improve in the paper.

**Essential References Not Discussed:**

The paper includes exhaustive references; however, the related work section is in the Appendix. Since the paper re-states some of the claims in existing papers, it would be better to include at least one paragraph of related work in the main paper.

**Experimental Designs Or Analyses:**

All the experimental settings look sound.

**Methods And Evaluation Criteria:**

Yes.

**Other Comments Or Suggestions:**

1. Line 873: The expression for $p_i$ does not make sense. It should be $p_i = \sum_{j=1}^L \delta(x^j = i) / L$.

2. It might be good to show some decoding trajectories on the Sudoku or Zebra puzzles where vanilla unmasking makes mistakes whereas the adaptive strategy circumvents them?

**Other Strengths And Weaknesses:**

The contribution of the paper is incremental. It combines claims from existing papers (Nie et al. (2024) and Zheng et al. (2024)). That said, since the paper only focuses on one claim, it is easy to read and follow.

**Questions For Authors:**

1. Do you observe any difference between top-k and top-k margin-based samping on text MDMs? Can you provide some examples of the generated text for various sampling strategies?

**Relation To Broader Scientific Literature:**

As mentioned above, Nie et. al. (2024) and Zheng et. al. (2024b) together have already demonstrated the main claim of this paper to a great extent. Nie et. al., demonstrated empirically on text data that the complexity of training MDMs is higher than ARMs. That said, the scope of Nie et. al. was quite broad and was focused more on the scaling aspect of MDMs. Zheng et. al. introduced the top-k sampling strategy for MDMs and demonstrated that it works much better than random unmasking. However, Zheng et. al. do not discuss the learning aspect of MDMs. This paper is much narrower in scope and tries to tease out the essence of adaptive decoding for MDMs through theoretical arguments and carefully selected experiments.

**Theoretical Claims:**

The proof for Proposition 2.1 is correct. The claim in proposition 3.3 looks reasonable; however, I was unable to check the complete proof.

---

> ### Author Rebuttal · Authors · 2025-03-31
>
> We greatly appreciate the reviewer's overall positive evaluation and comments. We will make sure to include a paragraph of related work in the main body and fix the typo mentioned. Below, we respond to the reviewer’s main concerns:
>
> ## (1) Scope of our contributions
> The reviewer stated “The contribution of the paper is incremental. It combines claims from existing papers (Nie et al. (2024) and Zheng et al. (2024))” and  “Nie et. al. (2024) and Zheng et. al. (2024b) together have already demonstrated the main claim of this paper to a great extent.”
>
> We respectfully disagree with the reviewer on this point. As stated in our introduction, our goal is to **understand** the benefits and drawbacks of training and inference of MDMs over ARMs.
>
> - Even though Nie et al. (2024) show that the autoregressive models outperform MDMs in scaling, they don’t explain the reason behind it. In this work, we give extensive empirical and theoretical evidence that this is due to the **heterogeneity of complexity across masking tasks at training time** (Section 3). **While empirically it has been observed (even well before the work of Nie et al.) that MDMs are more difficult to scale, our paper is the first to provide rigorous insight into why this is the case.**
> - While Zheng et al. (2024) propose a different ordering of the sampling, we view our most important contributions in this direction not to be about proposing new heuristics per se, but about explaining the reason/motivations behind the improvement achieved by these heuristics. Indeed, in our work we **provide principled justification for such adaptive inference schemes**, e.g. by showing that *any-order inference in perfectly trained MDM results in the same true distribution* (line 296, right column), and **disentangle the extent to which different “confidence-based” decoding strategies are actually planning based on uncertainty.**
> - Additionally, both of these works fail to explain that the benefit of MDM (especially with adaptive inference) over ARMs is most dramatic on tasks where the left-to-right token ordering structure *doesn’t hold*. This also explains the reason behind very drastic improvements in tasks like math or coding (e.g., see Table 1 in [1]) where left-to-right ordering doesn’t hold.
>
> [1] Large language diffusion models. Nie et al. 2025.
>
>
> ## (2) Top-k margin outperforms Top-k on challenging code and math tasks
> On text MDMs, for challenging math and coding tasks, we found that Top-k margin outperforms Top-k. For comparison, we adapted LLaDA [1], 8B MDM. For the result, please refer to our response to the reviewer xjPE. Notably, in more challenging tasks, such as HumanEval-Multiline, HumanEval-Split Line, and Math, **Top-k Margin shows a clear advantage over Top-k**.
>
> This is because the Top-k margin offers a more reliable estimate of uncertainty when multiple tokens have similar probabilities (our claim in Section 4.1)—a common scenario in the challenging tasks in the coding and math domains. **These results also further highlight the potential of the Top-k Margin strategy for challenging infilling tasks.**
>
> To understand the difference between Top-K and Top-K margin strategy, we consider the following problem.
>
> The problem prompt given to an MDM is: [If $\sqrt{5x}\cdot\sqrt{10x}\cdot\sqrt{18x}=30$, find $x$.]
> The model’s output using Top-K strategy is:
>
>
> …
> *So the equation becomes:
> \[
> \sqrt{900x^3} = 30
> \]*
>
> *Square both sides to eliminate the square root:
> \[
> (900x^3)^2 = 30^2
> \]*
> ….
> The model was wrong by decoding an incorrect sentence (900x^3)^2 = 30^2. At the moment just before decoding ^ (following 900x^3), the model faces multiple plausible options: (1) adding ^, or (2) adding =. This ambiguity arises from the token "square", which confuses the model. The Top-k strategy selects ^, as it has the highest probability. **This exemplifies a situation where the model assigns comparable probabilities to multiple tokens at a single location**. In contrast, the probability margin between ^ and = was small, indicating high uncertainty, so Top-k margin shifted focus to a different position where it had greater confidence, leading to the correct statement, *900x^3 = 900*.
>
> ## (3) Examples of decoding trajectories for Sudoku:
> For the following partial Sudoku board, Vanilla MDM inference decodes a cell at random—for example, the cell in the 7th row and 9th column. In contrast, adaptive MDM inference prioritizes cells in the 1st and 2nd rows, which are objectively easier to fill in at earlier.
>
> |   |   |   |   |   |   |   |   |   |
> |---|---|---|---|---|---|---|---|---|
> | 9 | 8 | 3 | 7 | 5 | . | 4 | 1 | 2 |
> | 2 | 4 | 5 | . | 9 | 1 | 3 | . | 6 |
> | 7 | 1 | 6 | 2 | 3 | . | . | . | . |
> | . | 2 | 1 | . | . | 8 | . | . | . |
> | 3 | 7 | . | . | 1 | . | 2 | 6 | . |
> | 6 | 9 | . | . | 2 | . | 8 | . | 1 |
> | 8 | . | 2 | . | . | . | 1 | . | . |
> | 1 | 5 | 7 | . | 8 | 3 | 6 | . | . |
> | . | 6 | . | 1 | 7 | . | 5 | 3 | 8 |

---

### Official Review · Reviewer_xjpE · 2025-03-14

**Overall Recommendation:** 5

**Summary:**

This work presents two contributions to a emerging discrete diffusion model called masked diffusion models.

* The first contribution is a theoretic construction showing the hardness of prediction subtasks within masked diffusion, motivating an inference time solution to sidestep these challenging subtasks.

* The authors then proposed a new criteria to adaptively choose the decoding order in inference time based on probability margins and show that leads to significant performance boosts of masked diffusion models in planning problems.

**Claims And Evidence:**

Yes

**Essential References Not Discussed:**

Relevant work in literature is well cited.

**Experimental Designs Or Analyses:**

Yes. The experiment design is thoughtful and closely tracks the main claims.

**Methods And Evaluation Criteria:**

Yes

**Other Comments Or Suggestions:**

Can you test your decoding strategy on common text or image tasks often used to evaluate diffusion models?

**Other Strengths And Weaknesses:**

## Strengths

* The theoretical results on provable hardness of mask prediction problems in some orders is original and improves our understanding of the intrinsic difficulty of training such models. It also provides sufficient motivation for the inference-time strategy that followed later.

* The results on planning tasks are pretty strong. The proposed probablity margin strategy significantly improves performance on hard Sudoku tasks. Notably, it even outperforms an AR model that is informed of the optimal order to solve the problem.

## Weaknesses

* The decoding-order selection, while novel, is still heuristic. The experimental evidence is in a narrow domain (e.g., sudoku puzzles) and the generality needs to be tested further, e.g., in text and image experiments.

* It's surprising and unclear why the masked diffusion model that uniformly optimizes predictions over all ordering can ourperform AR models informed of the optimal ordering. Could the authors elaborate on the possible reasons and implications?

**Questions For Authors:**

N/A

**Relation To Broader Scientific Literature:**

This paper improves the understanding of masked diffusion models - presenting theoretical analysis of the hardness of the prediction problems, which motivates inference-time improvements that selects "easier" decoding passes.

**Theoretical Claims:**

Yes. I have followed the presented theoretical results, including proposition 2.1 and the tractability of subtasks in Example 3.2.

---

> ### Author Rebuttal · Authors · 2025-03-31
>
> We greatly appreciate the reviewer's positive evaluation and the insightful comments and questions. Below, we respond to the reviewer’s suggestions and questions.
>
> ## (1) Further experiments on text data--Top-k margin outperforms Top-k on challenging code and math tasks
>
> To examine the effect of different inference strategies on text evaluation tasks, we adapted LLaDA, the 8B MDM model from [1]. We compare three strategies: **Vanilla, Topk, Topk prob -margin**.  The results are presented below.
>
> | **Sampler**      | HumanEval-Single | HumanEval-Multi | HumanEval-Split | Math        | MMLU-Pro | ROCStories |
> |------------------|-------------------------|------------------------|------------------------|--------|----------|-------------|
> | **Vanilla**      | 31.8%                   | 16.5%                  | 14.2%                  | 28.5%  | 33.2%    | 21.23%      |
> | **Top-k**        | 32.9%                   | 20.8%                  | 18.4%                  | 31.3%  | **36.5%** | 21.10%      |
> | **Top-k Margin** | **33.5%**               | **25.4%**              | **22.3%**              | **34.3%** | 35.4%    | **21.41%**   |
>
>
> As shown in the table, both Top-k and Top-k Prob. Margin consistently outperform vanilla MDM inference, underscoring the importance of adaptively selecting the decoding order to avoid harder problem instances. Notably, in relatively challenging tasks, such as HumanEval-Multiline, HumanEval-Split Line, and Math, **Top-k Margin shows a clear advantage over Top-k**.
> This is because the Top-k prob margin offers a more reliable estimate of uncertainty when multiple tokens have similar probabilities—a common scenario in these challenging tasks. In addition, particularly in coding and math problems where often a fixed answer exists, selecting the correct intermediate token during inference is critical. (Hence, a wrong token selection can directly lead to the incorrect answer) These results reinforce our claim in Section 4.1: Top-k Margin serves as a better proxy for positional uncertainty than Top-k in such cases. **These results also further highlight the potential of the Top-k Margin strategy for challenging infilling tasks.**
>
>  ## (2) On the reason why MDMs outperform ARMs
> MDM is trained across all possible orderings and uses an adaptive inference strategy. This flexibility allows it to discover more efficient reasoning orders tailored to the task or dataset, which can generalize better to unseen data. In contrast, ARMs trained with a fixed order may fail to generalize to unseen (or harder) data (please refer to Table 4 in our paper). Additionally, the harder MDM training (i.e., training in more than one token generation order) might be more (sample) efficient than ARM training that focuses on learning in only one order. Moreover, the ordering used for ARM training—predetermined by humans—may be suboptimal. In contrast, MDM may discover more effective decoding orders by systematically leveraging information from the logits, often outperforming human-specified orderings.
>
> ## (3) Implications
> These hint at the strong potential of MDMs, which we also highlighted in Section 4.3: Since MDMs are trained on all possible masked subproblems, their adaptive inference allows them to discover *good reasoning paths*, potentially leading to better performance than fixed orderings predetermined by humans.
>
> [1] Large language diffusion models. Nie et al. 2025

---

### Decision · Program_Chairs · 2025-05-01

**Decision:**

Accept (oral)

**Comment:**

The paper theoretically argues that masked diffusion models effectively solve a harder problem than autoregressive models. However, they show that this can be circumvented if one were to adaptively order the token decoding. This is emprically demonstrated on logic (Sudoku) puzzles (and later in the authors' response, Math and HumanEval-coding tasks).


All reviewers agree that the results here improve our understanding of masked diffusion models significantly (`xjpE`, `qGCY`), the theory is original (`xjpE`) and overall findings maybe useful in building future diffusion models (`oBo5`). The reviewers agree that the experiments are  well-designed (`xjpE`, `XERQ`) and sound (`qGCY`) and support the claims (`xjpE`). Reviewers (`XERQ`, `oBo5`) say the proofs look correct or reasonable.

Reviewers `xjpE` and `oBo5` also say that the results are strong as far as Sudoku is concerned; however multiple reviewers raised the concern that there were no gains in Zebra, and no results were presented in text/image tasks (`xjpE`, `qGCY`, `oBo5`). The authors have responded with new affirmative results on math/coding tasks. I believe this addresses the reviewers concerns adequately.

As Reviewer `qGCY` suggests in detail, in future versions of the paper, I hope that the authors will take care to make their claims more precise.